# Nitrate-functionalized patch confers cardioprotection and improves heart repair after myocardial infarction via local nitric oxide delivery

Dashuai Zhu [1,2,10], Jingli Hou[3,10], Meng Qian[1,10], Dawei Jin[4], Tian Hao[1], Yanjun Pan[4], He Wang[1], Shuting Wu[4], Shuo Liu [2], Fei Wang[1], Lanping Wu[5], Yumin Zhong[6], Zhilu Yang[7], Yongzhe Che[2], Jie Shen[8], Deling Kong[1], Meng Yin [4 ✉] & Qiang Zhao [1,9 ✉]

Nitric oxide (NO) is a short-lived signaling molecule that plays a pivotal role in cardiovascular system. Organic nitrates represent a class of NO-donating drugs for treating coronary artery diseases, acting through the vasodilation of systemic vasculature that often leads to adverse effects. Herein, we design a nitrate-functionalized patch, wherein the nitrate pharmacological functional groups are covalently bound to biodegradable polymers, thus transforming small-molecule drugs into therapeutic biomaterials. When implanted onto the myocardium, the patch releases NO locally through a stepwise biotransformation, and NO generation is remarkably enhanced in infarcted myocardium because of the ischemic microenvironment, which gives rise to mitochondrial-targeted cardioprotection as well as enhanced cardiac repair. The therapeutic efficacy is further confirmed in a clinically relevant porcine model of myocardial infarction. All these results support the translational potential of this functional patch for treating ischemic heart disease by therapeutic mechanisms different from conventional organic nitrate drugs.

[1] State key Laboratory of Medicinal Chemical Biology, Key Laboratory of Bioactive Materials (Ministry of Education), College of Life Sciences, Nankai University, Tianjin, China. [2] School of Medicine, Nankai University, Tianjin, China. [3] Tianjin Key Laboratory on Technologies Enabling Development of Clinical Therapeutics and Diagnostics, School of Pharmacy, Tianjin Medical University, Tianjin, China. [4] Department of Cardiothoracic Surgery, Shanghai Children's Medical Center, School of Medicine, Shanghai Jiao Tong University, Shanghai, China. [5] Department of Cardiac Ultrasound, Shanghai Children's Medical Center, School of Medicine, Shanghai Jiao Tong University, Shanghai, China. [6] Diagnostic Imaging Center, Shanghai Children's Medical Center, School of Medicine, Shanghai Jiao Tong University, Shanghai, China. [7] Key Laboratory of Advanced Technology for Materials of Education Ministry, School of Materials Science and Engineering, Southwest Jiaotong University, Chengdu, China. [8] College of Pharmacy, Nankai University, Tianjin, China. [9] Zhengzhou Cardiovascular Hospital and 7th People's Hospital of Zhengzhou, Zhengzhou, Henan Province, China. [10] These authors contributed equally: Dashuai Zhu, Jingli Hou, Meng Qian. ✉email: yinmengmdphd@163.com; qiangzhao@nankai.edu.cn

Cardiovascular disease (CVD) is a leading cause of mortality and morbidity worldwide, and ischemic heart disease accounts for approximately half of all deaths[1]. Coronary artery obstruction results in the occurrence of myocardial ischemia, which causes the death of non-regenerative cardiomyocytes and the subsequent formation of fibrotic scars. Strategies for the treatment of myocardial infarction (MI) in the clinic involve vasodilation and antiplatelet therapeutics[2,3], among which organic nitrate derivatives are the first choice and have been utilized for more than a century. The therapeutic effects of organic nitrates have been ascribed to the bioactive molecule nitric oxide, which is generated as a metabolic product through both enzymatic and non-enzymatic pathways[4].

As a versatile signaling molecule, nitric oxide (NO) plays critical roles in regulating cardiovascular homeostasis. Dysfunction of the NO signaling pathway is always correlated with the increased morbidity of MI[5]. In this regard, exogenous supplementation with NO can prevent infarction formation[6–8] by relaxing vascular tone, inhibiting platelet aggregation[9,10], and modulating the inflammatory response[11], adding a cardioprotective effect[12].

Recently, significant progress has been made in the development of innovative therapeutic strategies for MI, including both cell-based[13,14] and biomaterial-based[15–18] approaches, and some of them are undergoing pre-clinical or clinical trials[19]. Cardiac patches represent an important type of strategy that has received increasing attention for the treatment of the ischemic myocardial injury. Cardiac muscle patches of clinically relevant sizes have been successfully developed from human induced-pluripotent stem cells (hiPSCs) by different research groups[20,21]. Acellular cardiac patches are another kind of therapy that treat MI by restricting ventricular dilatation, preventing adverse left ventricular remodeling, and improving contractile function that is directly attributable to the elastic (such as polyurethane) or viscoelastic properties[22,23]. In addition, biomaterial-based patches can also act as carriers to deliver biological moieties with therapeutic functions, including miRNA, proteins, and so on refs. [24,25].

In the present study, we designed a NO cardiac patch based on an original and different concept, wherein the nitrate moiety was covalently bound to biodegradable poly(ε-caprolactone) (PCL), thus transforming small-molecule drugs into functional biomaterials. The biodegradability and biocompatibility of the material guarantees that this patch can be directly implanted onto the heart, and the implanted patch demonstrated local NO delivery to infarcted myocardium that is controlled by the ischemic microenvironment, a vast improvement and advantage over glyceryl trinitrate (GTN) patches. Site-specific delivery of NO provided effective cardioprotection, thus markedly ameliorating heart function and attenuating adverse remodeling. The therapeutics efficacy was further confirmed in a clinical-relevant porcine MI model, with the intent to establish the substantial clinical viability and improvement of the NO cardiac patches.

## Results

**Fabrication and characterization of nitrate-functionalized cardiac patches**. PCL oligomers with both ends capped with nitrates (PCL-ONO$_2$) were first synthesized by acylation of PCL-diol ($M_n = 2000$) with 4-bromobutanoyl chloride, followed by substitution of the terminal bromide with AgNO$_3$ (Supplementary Fig. 1), and the structure of PCL-Br and PCL-ONO$_2$ was verified by $^1$H NMR and $^{13}$C NMR spectra (Supplementary Figs. 2, 3). Then, PCL-ONO$_2$ was blended with high molecular weight PCL ($M_n = 80,000$) at a ratio of 1:9 (m/m) and processed into a fibrous mat by electrospinning (Fig. 1a). As a control, plain PCL patches were prepared by the same protocol. The scanning

electron microscope (SEM) image showed a homogenous fibrous structure with an average fiber diameter of 0.69 μm and an average pore size of 3.4 μm, which is similar to those of the PCL patch (Fig. 1b and Supplementary Table 1). Mechanical properties were first characterized by tensile testing, and the NO patch demonstrated moderately decreased tensile strength and elongation rate due to the incorporation of low molecular weight PCL-ONO$_2$ (Fig. 1c and Supplementary Table 1). The addition of PCL-ONO$_2$ also improved surface hydrophilicity with a decreased contact angle (Fig. 1d).

When the patch was implanted subcutaneously in rats, the total NO$_x$ within the patch was reduced in a controlled manner, and the residual NOx exceeded 26.6% after 28 days of implantation (Supplementary Fig. 4).

**Local delivery and site-specific accumulation of NO in the infarcted myocardium**. A nitrate-functionalized cardiac patch was further utilized for the treatment of rat MI, and regional NO generation in the ischemic heart was first evaluated by electron paramagnetic resonance (EPR) assay using ferrous N-diethyl dithiocarbamate (Fe-DETC) as the spin-trapping reagent (Fig. 2a)[26]. The resultant NO adduct (DETC)$_2$Fe–NO exhibited a characteristic triplet EPR signal ($a_N = 12.78$ G, $g_{iso} = 2.041$) at room temperature (Fig. 2b). The quantitative data indicated that in the distal region, there was no detectable difference in NO levels among the various groups (Fig. 2c). In the infarcted myocardium, NO levels were moderately enhanced in both MI and MI + PCL groups compared to the sham group due to endogenous NO production by activated macrophages (Supplementary Fig. 5). The NO level was markedly elevated after NO patch treatment, which was significantly ($p < 0.001$) higher than that after PCL patch treatment (Fig. 2d), suggesting exogenous NO generation from NO patches in infarcted myocardium. As a comparison, the NO patch was also administered to the hearts of rats without MI; the NO level was much lower than that in NO patch-treated MI hearts (Fig. 2d). All these results support the augmented NO generation from nitrate-functionalized patches in the ischemic heart because of the hypoxic and acidic microenvironment that has been investigated extensively before[27]. Next, site-specific accumulation of NO in the infarcted myocardium was detected by near-infrared (NIR) fluorescence imaging using the NO-sensitive probe DAC-S, developed by Sasaki et al.[28] (Fig. 2e, f, Supplementary Figs. 6–8), and regional accumulation of NO was clearly demonstrated in the infarcted zone after NO patch treatment compared to the MI group without any treatment (Fig. 2f).

**Mechanism of biotransformation of nitrate-functionalized patch into NO**. The biotransformation of nitrate-functionalized patch into NO proceeds through a stepwise process (Fig. 3a), which is different from organic nitrates that directly liberate into vasculature and release NO via intracellular metabolism by specific enzymes. Nitrate-functionalized polymers cannot cross the cell membrane; instead, they first hydrolyze to release nitrate anion through a non-enzymatic process as demonstrated by an in vitro assay (Fig. 3b). Nitrate anion is the major hydrolysis product during early stage because of the relatively slow degradation of bulk PCL (Supplementary Fig 9). The released nitrate anion can further be metabolized to NO through the nitrate-nitrite-NO reduction pathway under the catalysis of various endogenous enzymes[29,30]. The conversion of nitrate and nitrite has been observed in PBS buffer with the addition of tissue homogenates from the hearts, which confirmed the presence of relevant reductases (Fig. 3c, d).

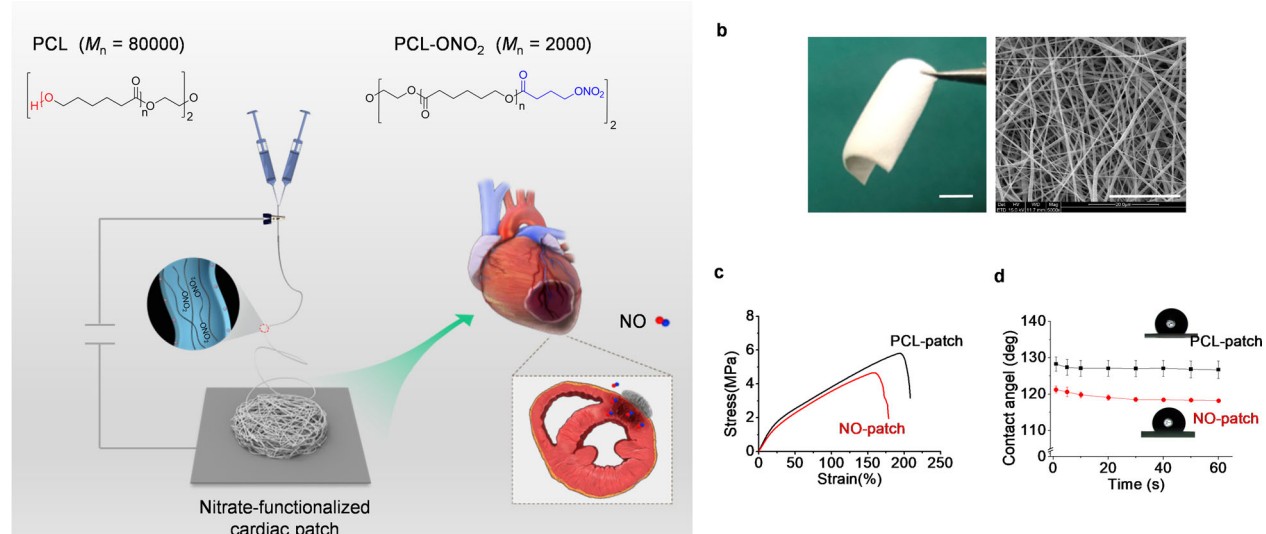

**Fig. 1 Fabrication and characterization of nitrate-functionalized cardiac patch. a** Schematic illustration of the fabrication of nitrate-functionalized cardiac patch and application for the treatment of myocardial infarction. After blending PCL-ONO$_2$ with high molecular weight PCL, a nitrate-functionalized cardiac patch was fabricated through electrospinning, and implanted to cover the infarcted myocardium. Under ischemic microenvironment, nitric oxide was generated from the nitrate-functionalized patch to offer therapeutics for MI. **b** Representative images of the nitrate-functionalized cardiac patch (scale bar, 5 mm) and its microstructure demonstrated by SEM (scale bar, 20 μm). Three independent tests were repeated with similar results. Characterization of the nitrate-functionalized patch in terms of mechanical properties (**c**) and water contact angle (**d**). Data are expressed as mean ± SD, $n = 3$ independent repeats.

More importantly, the step-by-step transformation is responsive to the pathologic microenvironment of the ischemic heart. First, hypoxia occurred due to the obstruction of blood flow. Following left anterior descending (LAD) ligation, cardiac partial pressure of oxygen (pO$_2$) decreased (Fig. 3e, Supplementary Fig 10), indicating obvious hypoxia. In addition, ischemia also leads to the local accumulation of H$^+$ with pH of about 5.7 as determined by $^{31}$P NMR (Fig. 3f, Supplementary Fig. 11). As a result, acidotic microenvironment accelerates the non-enzymatic hydrolysis of nitrate-functionalized PCL (Step 1), which is a crucial step in determining the whole process of NO biotransformation (Fig. 3b). The conversion of nitrate and nitrite by the reductase within the heart tissue has also been enhanced under the acidotic and reduced conditions as reported before[27].

In addition, the local and sustained generation of NO by the nitrate-functionalized patches efficiently abolishes side effects, such as hypotension observed in the utilization of organic nitrate drugs by oral or transdermal delivery route due to systemic NO delivery (Supplementary Fig. 12)[31,32].

**Local NO delivery provides a protective effect in cardiac ischemia**. Nagar-Olsen staining was performed to detect cardiac injury at an early stage, in which the damaged myocardium was stained red. After NO patch treatment, the staining area was significantly diminished, suggesting relieved cardiac injury (Fig. 4a). Cell apoptosis at both border and infarcted zones was also decreased, as illustrated by terminal deoxynucleotidyl transferase dUTP nick end labeling (TUNEL) staining (Fig. 4b, Supplementary Fig. 13). These data support the cardioprotection conferred by the NO patch in MI.

To further explore the underlying mechanism, myocardial blood flow (MBF) was evaluated through intraventricular perfusion of fluorescent microspheres[33]. Treatment with the NO patch after LAD artery ligation efficiently restored blood perfusion in the border zone in contrast to treatment with the PCL patch (Fig. 4c and Supplementary Fig. 14). Moreover, implantation of NO patches elevated pO$_2$ levels in MI hearts,

suggesting improved oxygen distribution and perfusion status. Implantation of NO patches to non-infarcted hearts resulted in negligible changes to pO$_2$ in the myocardium (Supplementary Fig. 10).

The cardioprotective function of NO has been attributed to the vital role it plays in the mitochondrial respiratory[12]. Hence, we detected the activities of mitochondrial complex I/II. We found that both the activities of mitochondrial complexes I and II were suppressed after NO patch treatment (Fig. 4d), and consequently, the level of hydrogen peroxide (H$_2$O$_2$) in cardiac tissues was reduced (Fig. 4d). Endogenous NO may cause S-nitrosylation (SNO), a type of posttranslational modification that can regulate protein function and cellular signaling. It has been accepted that the enhanced SNO of mitochondrial complex I inhibits its activity, thereby limiting ROS generation and providing cardioprotection[12,34,35]. In this study, we also observed that treatment with NO-patch increased the level of S-nitrosylated proteins in MI heart (Supplementary Fig 15). These results implied that NO delivery by NO patches quenched ROS production from mitochondria, which was beneficial for reducing cell apoptosis and ameliorating cardiac injury.

**Local NO delivery modulates cardiac inflammation**. The inflammatory response after patch implantation was first evaluated by histology analysis; haematoxylin and eosin (H&E) staining demonstrated decreased leukocyte infiltration after NO patch treatment compared to PCL patch treatment 3 days post-surgery (Supplementary Fig. 16). In addition, robust cardiomyocyte survival in the cardiac tissues was detected in NO patch-treated hearts. Macrophage infiltration in the infarcted hearts was further investigated by immunostaining with CD68, and the number of infiltrated macrophages was significantly ($p < 0.001$) decreased in the NO patch-treated hearts compared to the MI and PCL patch-treated hearts (Fig. 5a).

The modulatory effect of local NO delivery on macrophage polarization was further evaluated by immunostaining targeting CD86 and CD206, markers of M1 macrophages and M2

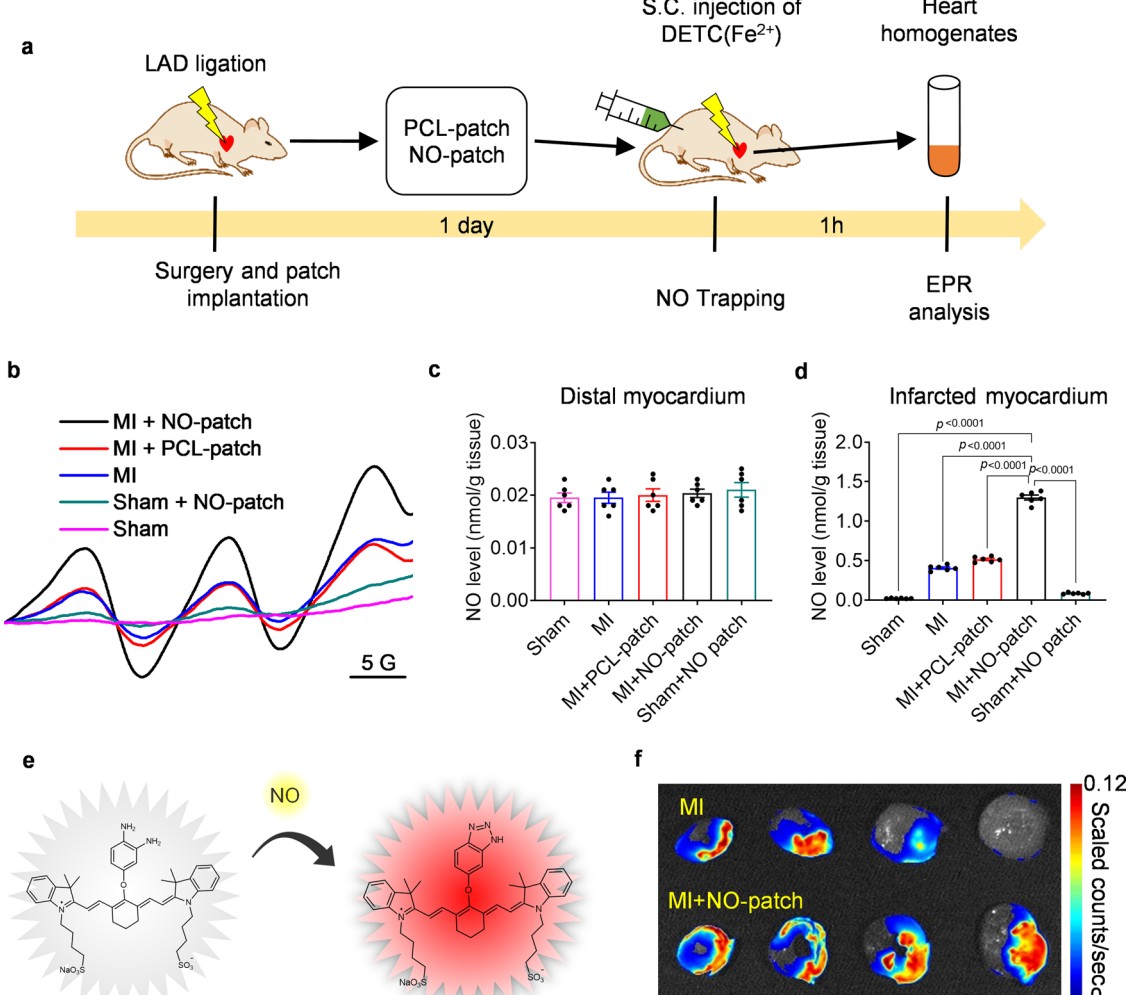

**Fig. 2 Enhanced NO generation in the infarcted myocardium because of hypoxic and acidic microenvironments. a–d** NO-patch was applied for the treatment of MI in rats, and NO accumulation in the myocardium was measured by EPR assay. One day after surgery, cardiac tissues were collected and divided into infarcted part and distal part, which were subjected to EPR analysis, respectively. **a** Schematic illustration of EPR analyses. **b** Representative EPR spectra reflecting NO generation in the infarcted myocardium in the presence of (DETC)$_2$Fe. NO levels in the distal (**c**) and infarcted (**d**) myocardium were determined by quantitation of (DETC)$_2$Fe–NO complex using 2,2,5,5-tetramethyl piperidine 1-oxyl (TEMPO). Data are expressed as mean ± SEM, $n = 6$ animals for each group. Significant differences were detected by two-tailed one-way ANOVA with Tukey's multiple comparisons test. **e, f** NO accumulation in the myocardium was detected by imaging of the infracted myocardium with NO-sensitive fluorescent probe. The principle for the detection of NO by the probe was shown in (**e**). The probe DAC-S was non-fluorescent; NO induced a significant fluorescence turn-on response at 805 nm. Representative NIR fluorescence images of infarcted myocardium in different locations were shown in (**f**). LAD: left anterior descending artery; EPR: electron paramagnetic resonance; DETC(Fe$^{2+}$): ferrous N-diethyl dithiocarbamate; NO: Nitric oxide.

macrophages, respectively. The results showed that after NO patch treatment, the number of M2 (anti-inflammatory and reparative) macrophages was significantly increased, while the quantity of M1 (pro-inflammatory) macrophages was decreased (Fig. 5b, c). Moreover, the expression of IL-1β and TNF-α, two classic pro-inflammatory cytokines of M1 macrophages, was also decreased, while the expression of Arg1 and IL-10, anti-inflammatory cytokines of M2 macrophages, was significantly upregulated after NO patch administration (Fig. 5d). Collectively, local NO delivery demonstrated a modulatory effect on inducing macrophages into the M2 phenotype, thus prohibiting inflammation and promoting tissue repair and regeneration.

**Local NO delivery improves cardiac function in MI rats.** Echocardiography assessment demonstrated that the NO patch effectively improved cardiac function with enhanced values of ejection fraction (EF) and fractional shortening (FS) 1-day

post-surgery, and the improvement lasted over 4 weeks. In contrast, the PCL patch showed only a temporary supportive effect (1 day) that was eventually diminished at 28 days; both the EF and FS were similar to those in MI rats, which exhibited obvious implications of heart failure (Fig. 6a). Consistently, the increased dimensions of the left ventricle (LVIDd and EDV) in the MI and PCL patch-treated groups also indicated the occurrence of dilated cardiomyopathy, one common cause of congestive heart failure (Fig. 6a).

The long-term effect of the NO patch on cardiac remodeling after ischemic injury was further investigated by histological analysis. Masson trichrome staining showed that treatment with the NO patch significantly ($p < 0.01$) reduced the infarct size and sustained the thickness of the interventricular septum (IVS). Concurrently, evidently increased thickness of the infarcted left ventricular wall was observed in sections distal to the apex due to the increased cardiomyocyte survival in the infarcted zone

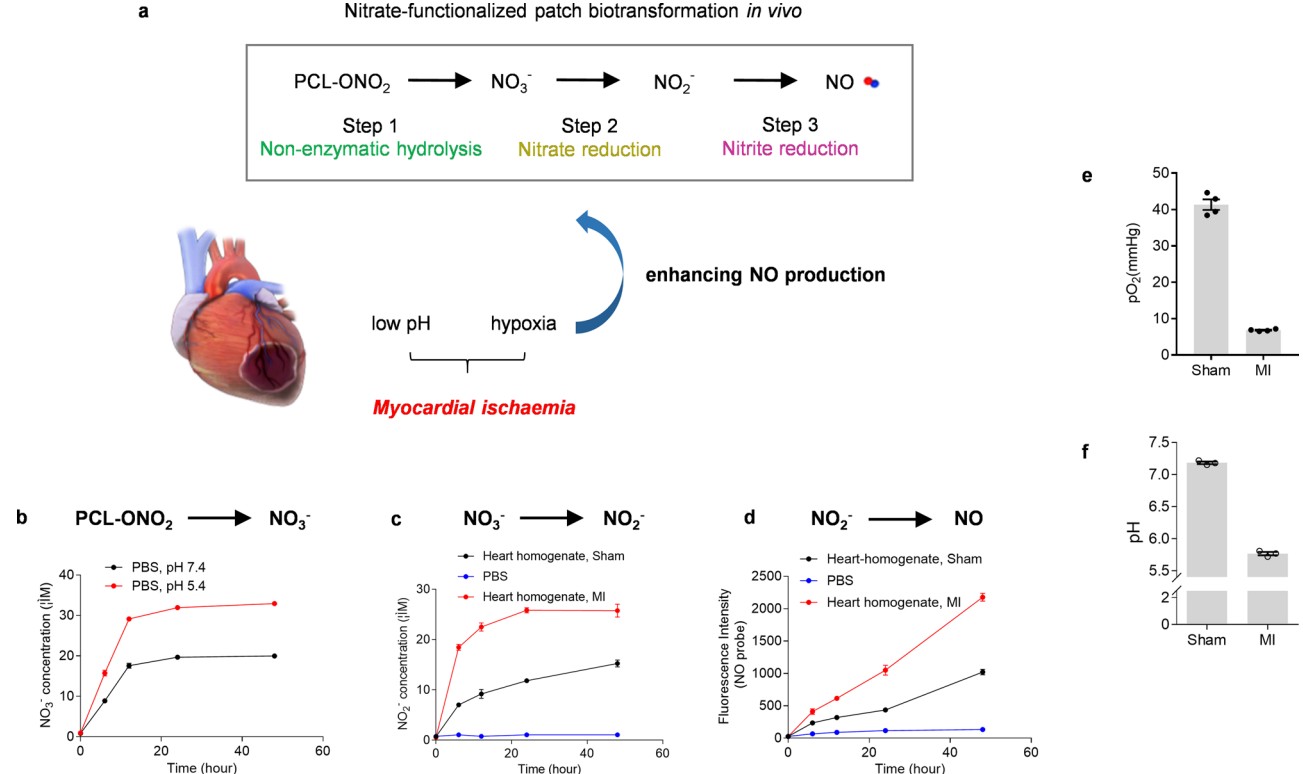

**Fig. 3 Mechanistic investigation of biotransformation of nitrate-functionalized patch in the ischemic heart. a** Schematic diagram illustrates the biotransformation pathway of $PCL-ONO_2$ into NO via a stepwise pathway. **b** In vitro release of $NO_3^-$ from nitrate-functionalized patch in PBS buffer (pH = 5.4 and 7.4) through the non-enzymatic hydrolysis of $PCL-ONO_2$ ($n = 3$). **c** Reduction of $NO_3^-$ into $NO_2^-$ in the presence of heart homogenate (10 mg/mL) ($n = 3$ independent repeats). **d** Reduction of $NO_2^-$ into NO in the presence of heart homogenate (10 mg/mL) ($n = 3$ independent repeats). **e, f** The levels of $pO_2$ ($n = 4$ animals per group) and pH ($n = 3$ animals each group) in sham operated or infarcted myocardium were measured. Data are expressed as mean ± SEM.

(Fig. 6b, c). The cross-section area of cardiomyocytes at the distal area of the infarction was further determined by WGA staining, and the NO patch group showed marked suppression of cardiomyocyte expansion compared to both the MI and PCL patch groups (Fig. 6d). Moreover, intensive collagen deposition was observed in MI and PCL patch-treated rats, showing typical pathological changes of heart failure (Fig. 6e).

Rebuilding of the vasculature favors regeneration and repair of the infarcted myocardium. In this study, local NO delivery effectively promoted angiogenesis with markedly enhanced capillary density at an early stage (day 3) (Fig. 6f). Neovascularization at 28 days was further evaluated by immunostaining targeting α-smooth muscle actin (α-SMA), a marker of functional arterioles; the number of α-SMA⁺ blood vessels distributed at the infarcted border zone was significantly ($p < 0.05$) higher in the NO patch group than in the MI and PCL patch groups (Fig. 6g).

To implement the research in a clinically applicable setting, we further introduced the NO-patch to a chronic model that underwent 4 weeks MI, and the therapeutics were evaluated (Fig. 7a). As shown by the results of Masson trichrome staining (Fig. 7b, c) and functional measurements (Fig. 7d, e), we concluded that implantation of NO-patch 4 weeks after MI effectively sustained cardiac function and suppressed cardiac dilation. These results further confirmed the benefit roles of NO-patch in a clinically applicable MI model.

**Local NO delivery improves cardiac function in a porcine model of MI.** The therapeutic efficacy of nitrate-functionalized cardiac patches was further evaluated in a clinically relevant porcine model of cardiac ischemia/reperfusion (I/R) injury (Fig. 8a, Supplementary Fig. 17a). First, the induction of MI was confirmed by electrocardiography (ECG) recording with an obvious ST-segment elevation after I/R (Supplementary Fig. 17b). At the same time, markedly increased levels of cardiac injury markers, serum levels of cardiac troponin I (cTnI) and Creatine kinase-MB (CK-MB), were detected (Supplementary Fig. 17c). Then, the NO patch was implanted to treat pigs with MI, and survival curves of MI pigs with or without NO patch treatment were plotted individually. I/R caused a mortality rate of 50% in the model group (I/R only), while the survival rate was efficiently enhanced after NO patch treatment (Fig. 8b). More importantly, the ejection and contraction functions were ameliorated, as demonstrated by the significantly ($p < 0.05$) enhanced EF and FS, together with decreased ESV and LVIDs 28 days post-surgery (Fig. 8c).

Cardiac magnetic resonance imaging (cMRI), a gold standard for accurate functional assessment in the clinic, was further performed to monitor cardiac performance and function. The results indicated marked enhancement in EF and cardiac output (CO) in NO patch-treated pigs compared to I/R injured pigs, confirming the improved cardiac function (Fig. 8d).

**Local NO delivery attenuates adverse remodeling in MI pigs.** To further elucidate tissue regeneration and remodeling after I/R injury, Masson trichrome staining was first performed in five sections (S1–S5) acquired from each heart (Fig. 9a), and accordingly, the fibrotic area in each section was measured (Fig. 9b). The area under the curve (AUC) was calculated to

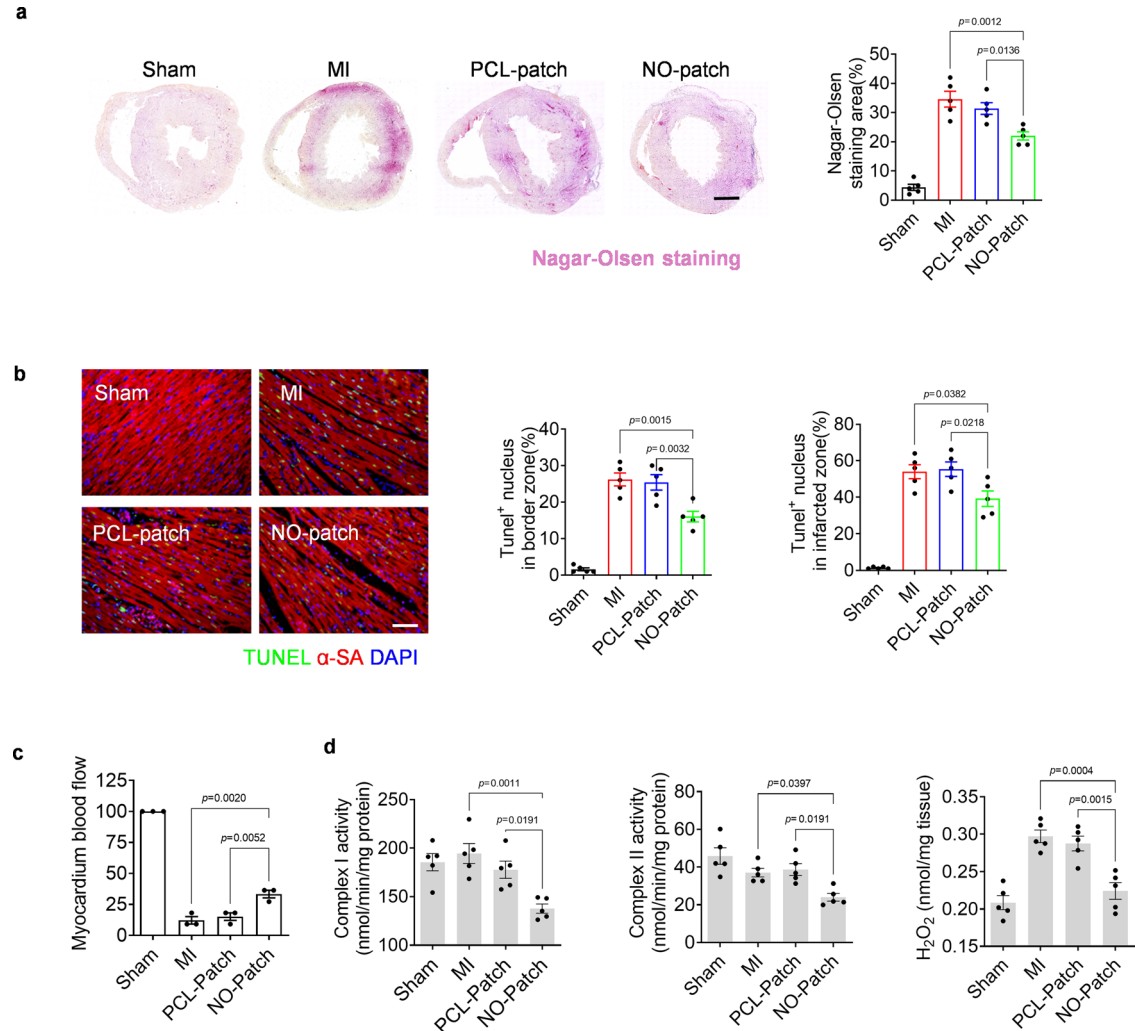

**Fig. 4 Local NO delivery by nitrate-functionalized cardiac patches confers cardioprotection against MI in rats at early stage (day 1). a** Representative Nagar-Olsen staining images showing ischemic damaged myocardium, which was stained in red (scale bar, 1 mm); the injury severity was further evaluated by measuring the staining area ($n = 5$ animals per group). **b** TUNEL staining was performed to detect apoptotic nucleus (scale bar, 40 μm), and the quantity of TUNEL positive nucleus in the ischemic border zone and infarcted zone was determined, respectively ($n = 5$ animals per group). **c** Relative myocardial blood flow was measured through intraventricular perfusion of fluorescent microspheres ($n = 3$ animals per group). **d** The activity of mitochondrial complex I and II as well as the hydrogen peroxide levels in cardiac tissues were measured one day post-surgery ($n = 5$ animals per group). Data are expressed as mean ± SEM. Significant differences were detected by two-tailed one-way ANOVA with Tukey's multiple comparisons test.

indicate the total scar size (Fig. 9c). From these results, we can learn that NO patch treatment strikingly decreased the infarct size. In addition, reduced infarct size was observed in NO patch-treated hearts through cMRI based on late gadolinium enhancement (LGE-cMRI) (Fig. 9d).

Next, neovascularization in the infarcted myocardium was detected by immunostaining for CD31 and α-SMA (Fig. 9e, f), and the results confirmed enhanced blood vessel regeneration (both capillary and arteriole) after NO treatment. Moreover, cardiomyocytes in the border zone of I/R injured hearts with NO patches also demonstrated pronounced Ki67 labeling (Fig. 9g), indicating increased proliferation by local delivery of nitric oxide.

## Discussion

Organic nitrates represent a conventional class of NO-donating agents that have been used to treat CVDs (ischemic heart disease, heart failure, and hypertension) for many years[4]. After uptake, organic nitrates first experienced denitration and were metabolized to nitrate ($NO_3^-$) or nitrite ($NO_2^-$) depending on the

structure of them[29]. The bioactivation is mediated by either non-enzymatic (endogenous reductants) or enzymatic reactions. Several enzymatic reactions are involved in GTN bioactivation, including aldehyde dehydrogenase 2 (ALDH2), xanthine oxidoreductase, and the cytochrome P450[36].

Currently, there is accumulating evidence demonstrating that inorganic nitrate represents an important reservoir for NO in the body[4,37]. Under specific conditions, inorganic nitrate can be metabolized to generate NO through the nitrate–nitrite–NO reduction pathway. Inorganic nitrate is found in considerable amounts in the diet, and many green vegetables such as spinach or beetroot are particularly rich in nitrate. Recently, supplementation of inorganic nitrate has been attempted by different groups with the aim of providing therapy for CVDs because it is free of tolerance phenomenon. Promising outcomes have been reported, including improved revascularization in chronic ischemia[38], and stabilized atherosclerotic plaques[39].

It has been widely accepted that nitrate can be reduced to nitrite mainly by commensal bacteria in the digestive system.

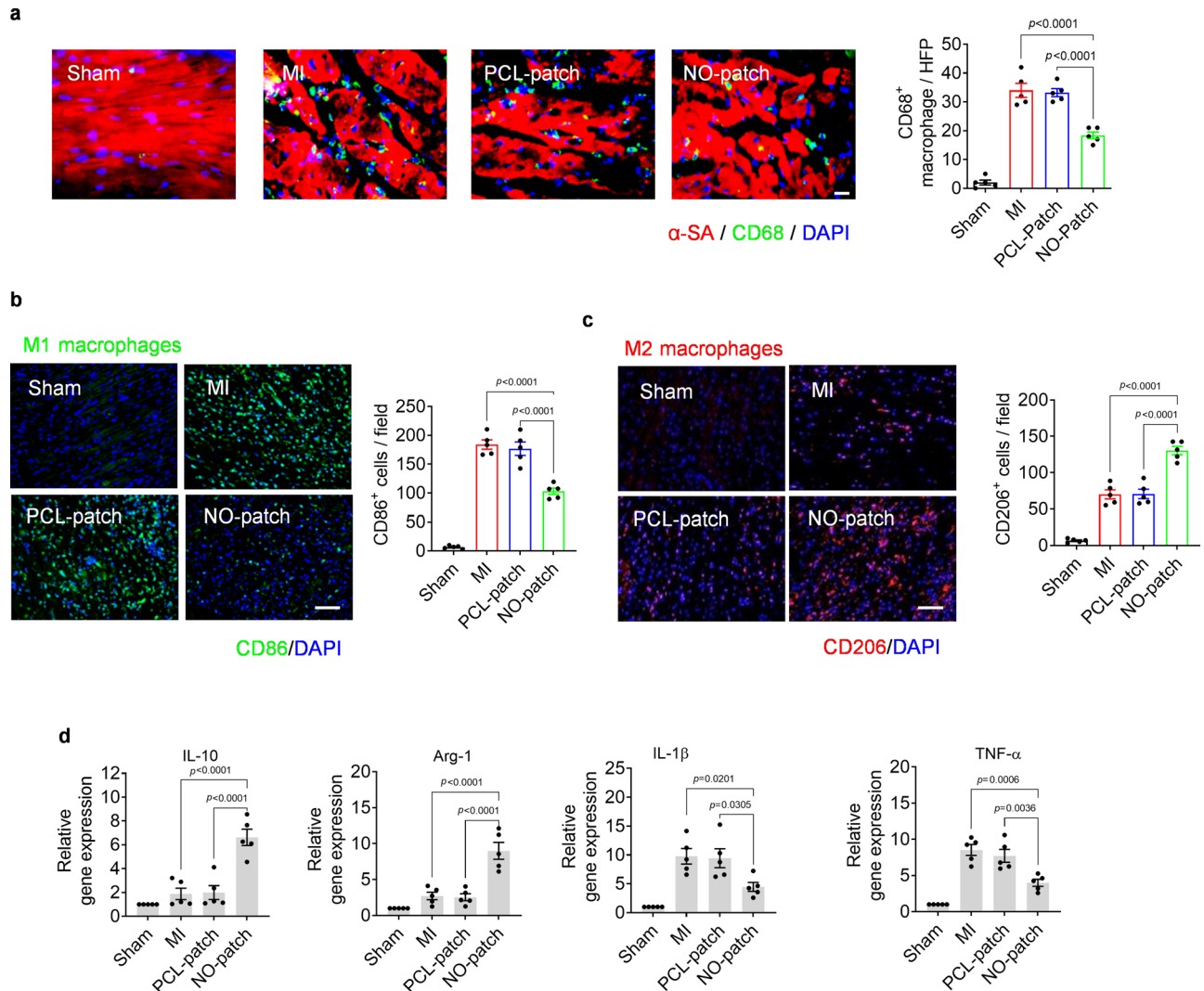

**Fig. 5 Local NO delivery by nitrate-functionalized cardiac patches modulates macrophage polarization in infarcted heart of rats at early stage (day 3).**
**a** Representative images showing macrophage infiltration in infarcted heart by immunostaining with CD68 (scale bar, 40 μm) as well as the corresponding quantitative analysis. **b**, **c** Macrophage polarization was further detected by immunostaining targeting CD86 and CD206, the maker of M1 and M2 phenotype, respectively (scale bar, 100 μm). **d** The expression of pro-inflammatory and anti-inflammatory cytokines was evaluated by RT-PCR. Data are expressed as mean ± SEM, $n = 5$ animals per group. Significant differences were detected by two-tailed one-way ANOVA with Tukey's multiple comparisons test. α-SA: α-sarcomeric actin; CD: cluster of differentiation; IL-10(interleukin-10); Arg-1(arginase-1); IL-1β(interleukin-1 β); TNF-α(tumor necrosis factor-α).

In addition, there are several recently discovered mammalian enzymes with nitrate reductase activity. Nitrite can then be further reduced to bioactive NO and consequently demonstrate direct vasoactive effects through various enzymatic pathways[40].

Among the different types of endogenous reductases for nitrite, myoglobin has received increasing attention in the cardiovascular system. It has been reported that deoxygenated myoglobin expressed in the heart can reduce nitrite to nitric oxide and thereby contribute to cardiomyocyte NO signaling during ischemia[41]. This effect has been confirmed in myoglobin knockout mice; that is, administration of nitrite reduced MI in myoglobin[+/+] mice with cardiac ischemia-reperfusion (I/R) injury, whereas the cardioprotective effect was abolished in myoglobin[−/−] mice[40]. Gladwin et al. also found that there is relatively little myoglobin present in lung tissue. Studies in the heart have revealed that 60% to 70% of NO generation from nitrite is mediated by the nitrite reductase activity of deoxymyoglobin, whereas in the lung, xanthine oxidoreductase (xOR) plays a role, with >70% of nitrite reduction mediated by xOR[42].

In this regard, nitrate and nitrite, previously considered inert oxidation products of NO metabolism, are now recognized as an important circulating reservoir of NO. On the other hand, in vivo transformation efficiency of inorganic nitrate is influenced by many factors, including the tissue, the pH, oxygen tension and redox status[43]. In addition, a major health concern is that higher doses of nitrite derived from dietary nitrate may cause severe methemoglobinemia. It may also lead to an increased cancer risk due to the formation of the carcinogen substances N-nitrosamines from ingested nitrate and nitrite[44]. As a result, the nitrate level in drinking water is strictly regulated in many countries[4].

In our study, EPR measurements clearly demonstrated that NO generation from nitrate-functionalized patches was ischemia-responsive; the NO signal in ischemic hearts was markedly elevated compared to that in non-ischemia hearts, with only marginal enhancement detected. This suggests that a low level of NO was generated from the nitrate-functionalized patch in a balanced physiological environment, whereas under ischemic conditions,

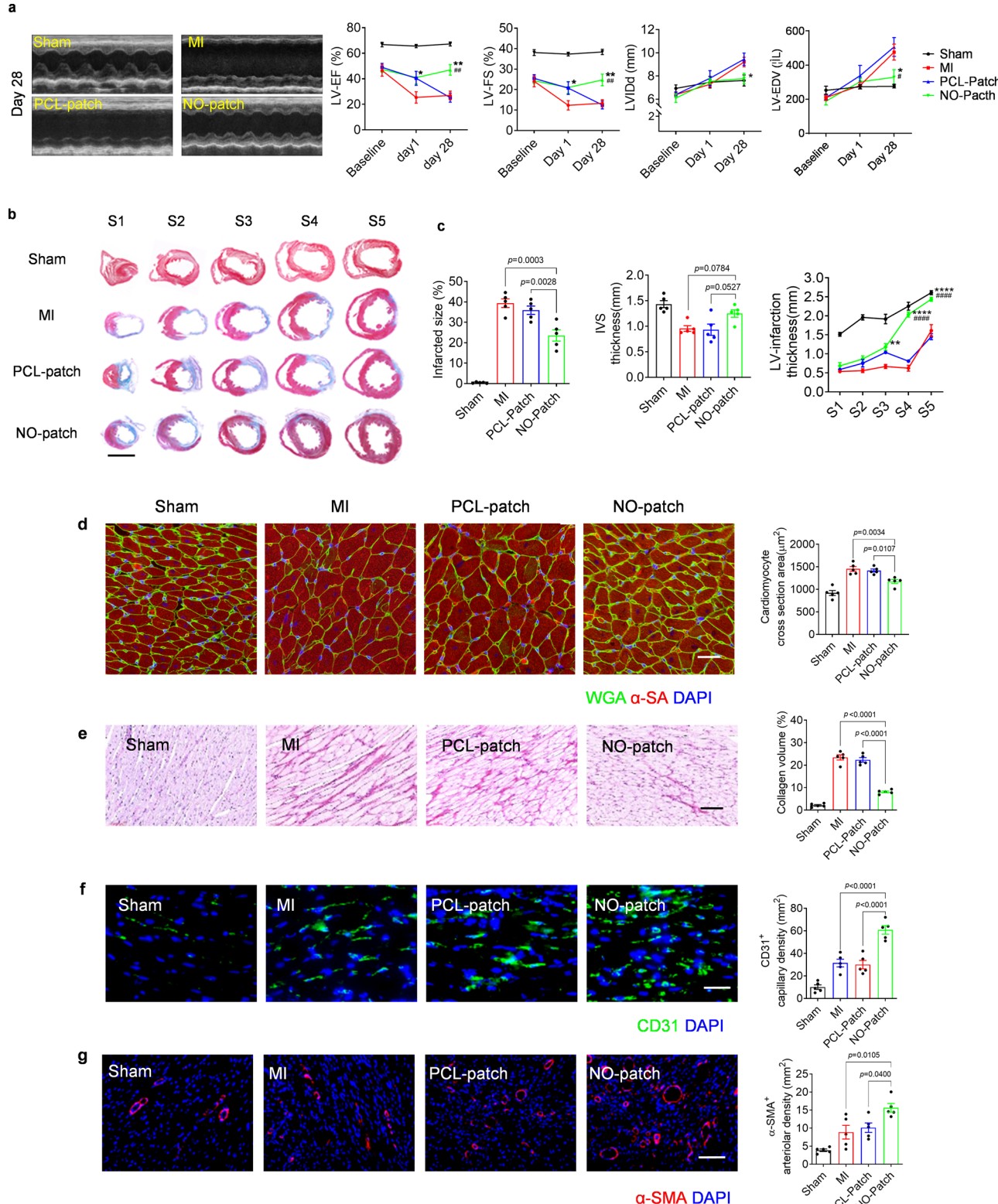

acidosis occurs as a result of anaerobic metabolism and $H^+$ accumulation[45,46] and to a greater extent with the disturbance in the redox that tends to be reductive because of the lack of oxygen species[47]. Besides, in situations of hypoxia, when oxygen-dependent NO synthases become dysfunctional, nitrite reduction is instead greatly enhanced[48]. These pathological events create the appropriate microenvironment that accelerates NO generation from nitrate-functionalized patches[27].

The underlying mechanism for the protective role of NO in cardiac ischemia has been extensively investigated during the past two decades[37]. Generally, the therapeutic efficacy can be explained from the following aspects.

First, mitochondria have been viewed as important targets in the cardioprotection provided by nitric oxide[49,50]. Nitric oxide (NO) has been known for many years to bind to cytochrome C oxidase, the terminal acceptor in the mitochondrial electron

**Fig. 6 Local NO delivery by nitrate-functionalized cardiac patches improves cardiac function and suppresses adverse cardiac remodeling in rats. a** Cardiac echo measurement was performed at different time points post-surgery, and cardiac function indicators of left ventricular-ejection fraction (LV-EF), left ventricular-fractional shortening (LV-FS), left ventricular internal diameter at end diastole (LVIDd) and left ventricular end-diastolic volume (LV-EDV) were evaluated accordingly. Data are expressed as mean ± SEM, $n = 5$ animals per group. Significant differences were detected by two-tailed two-way ANOVA with Tukey's multiple comparisons test. LV-EF: $p = 0.0254$ (Day 1, PCL patch vs MI), $p = 0.0199$ (Day 1, NO-patch vs MI), $p = 0.0017$ (Day 28, NO-patch vs MI), $p = 0.0006$ (Day 28, NO-patch vs PCL patch); LV-FS: $p = 0.0377$ (Day 1, PCL patch vs MI), $p = 0.0306$ (Day 1, NO-patch vs MI), $p = 0.0025$ (Day 28, NO-patch vs MI), $p = 0.0011$ (Day 28, NO-patch vs PCL patch); LVIDd: $p = 0.0271$ (Day 28, NO-patch vs PCL patch); LVEDV: $p = 0.0173$ (Day 28, NO-patch vs MI, $p = 0.0028$, NO-patch vs PCL patch). **b–g** Four weeks after surgery, the hearts were collected for histological analysis. **b** Masson trichrome staining was performed with five paraffin sections evenly separated from cardiac apex to base; infarcted size, thickness of infarcted left ventricular wall (LV-infarction) from S1–S5 and interventricular septum (IVS) were measured accordingly. Scale bar, 5 mm. **c** Data are expressed as mean ± SEM, $n = 5$ animals for each group. Significant differences were detected by two-tailed one-way or two-way ANOVA with Tukey's multiple comparisons test. **$p < 0.01$, ****$p < 0.0001$ vs MI, ####$p < 0.0001$ vs PCL patch. LV infarct thickness: $p = 0.0046$ (S3, PCL patch vs MI), $p < 0.0001$ (S3–S5, NO-patch vs MI; S4–S5, NO-patch vs PCL patch). **d** WGA staining to show cardiomyocyte hypertrophy that was characterized with increased cross-section area (scale bar = 40 μm). **e** Interstitial collagen deposition in the heart was detected by Sirius red staining (scale bar = 40 μm). **f** Hearts collected 3 days after surgery were evaluated by immunostaining with CD31, and CD31 positive capillaries were quantified. Scale bar, 60 μm. **g** Hearts collected 28 days after surgery were evaluated by immunostaining with α-SMA, and α-SMA positive arterioles were quantified. Scale bar, 60 μm. **d–g** Quantitative data are expressed as mean ± SEM, $n = 5$ animals for each group. Significant differences were detected by two-tailed one-way ANOVA with Tukey's multiple comparisons test.

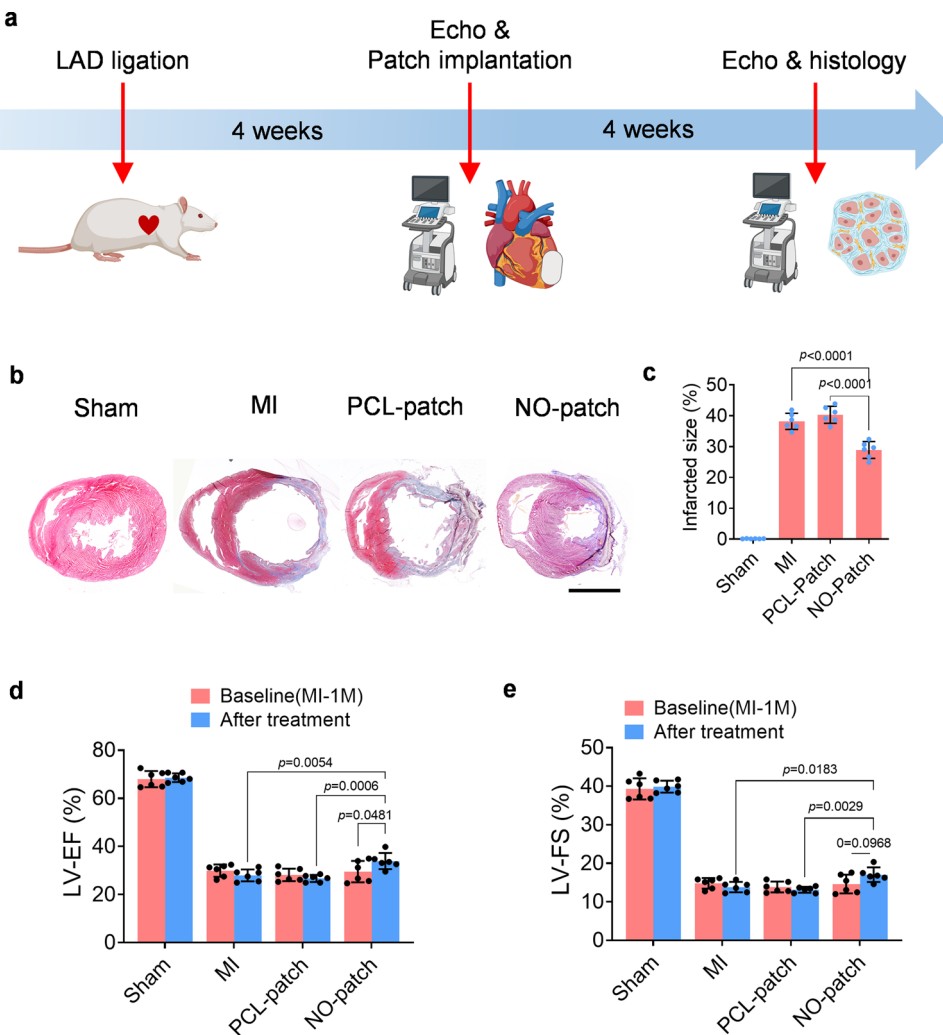

**Fig. 7 Local NO delivery exerts cardioprotective roles in a chronic rat MI model. a** Schematic image showing the NO-patch treatment for a chronic rat MI model. **b** Representative Masson trichrome staining images of the heart 4 weeks after different treatments. Scale bar, 5 mm. **c** The infarct size was calculated according to Masson staining. **d–e.** Quantitative data of echocardiography for measurement of cardiac function, and representatively, the left ventricular (LV) ejection fraction (LV-EF) and fraction shortening (LV-FS) were shown. Data are expressed as mean ± SEM, $n = 6$ animals for each group. Significant differences were detected by two-tailed one-way (**b**) or two-way (**d** and **e**) ANOVA with Tukey's multiple comparisons test. LAD: left anterior descending artery.

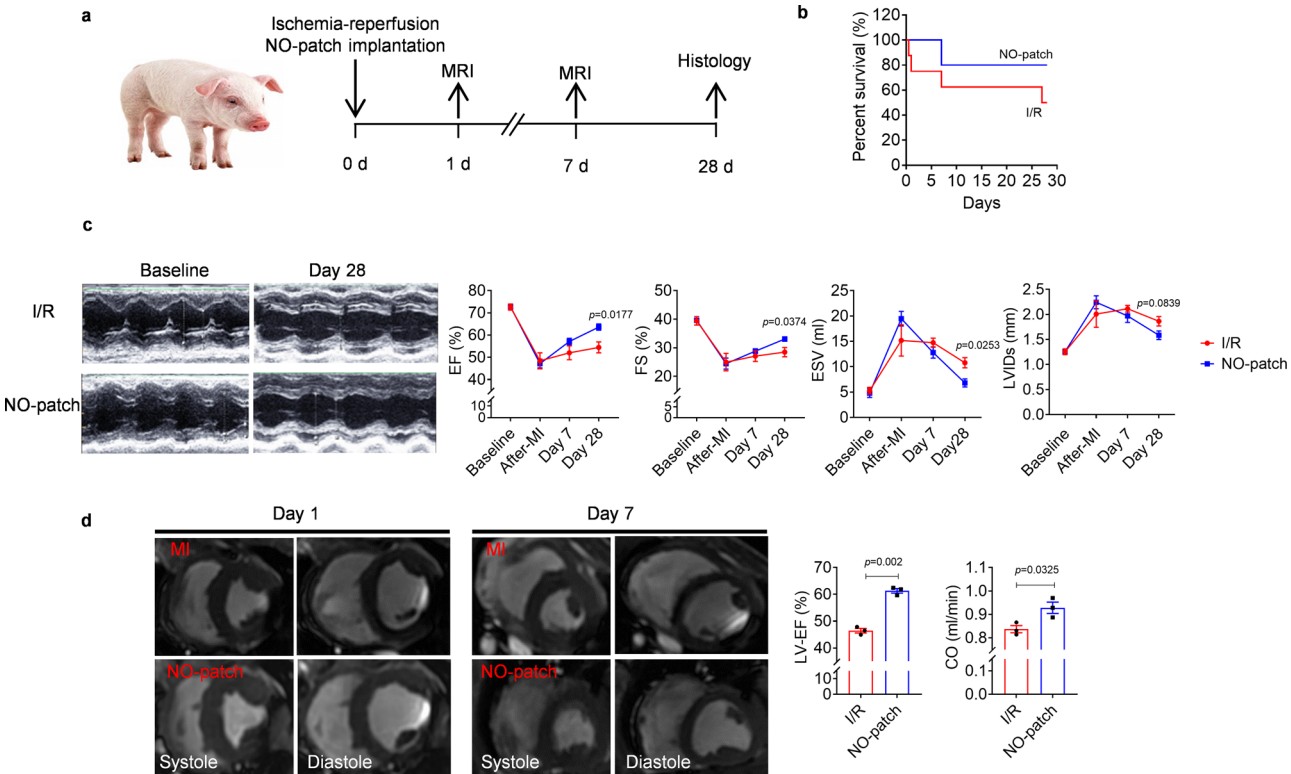

**Fig. 8 Local NO delivery by nitrate-functionalized cardiac patches enhances survival rate and improves cardiac function in MI pigs. a** Experimental schedule for the treatment of MI by epicardial implantation of NO patches. **b** Survival rates of pigs after MI with or without NO-patch treatment were recorded. NO patches improved survival of MI pigs. **c** Representative echo images reflecting cardiac function, and accordingly, left ventricular-ejection fractions (LV-EF), fraction shortening (LV-FS), left ventricular internal diameter at end diastole (LVIDs) and volume at diastole (ESV) were assessed ($n = 4$ pigs in I/R group, $n = 6$ pigs in NO-patch group). **d** Representative MRI images obtained on day 1 and 7 (left, end systole; right, end diastole) and LV-EF as well as cardiac output (CO) were measured on day 7 accordingly ($n = 3$ for each group). Data are expressed as mean ± SEM. Significant differences were detected by two-tailed student's $t$ test at indicated time points.

transport chain, in competition with oxygen[51]. This competition results in an inhibition of respiration that is more potent when oxygen tension is decreased.

The protective mechanism has been further elucidated by the fact that NO inhibits the activity of complex I in the mitochondrial electron transport chain through S-nitrosation of critical thiols on it, thereby limiting mitochondrial ROS generation[35]. This further prevents mitochondrial permeability transition pore opening and cytochrome c release[52]. Our results also demonstrated the inhibitory effect of NO on complex I as well as ROS generation. As a result, cardiomyocyte apoptosis was efficiently attenuated at an early stage in the nitrate patch-treated group. More prominent protective effects, including suppressed adverse cardiac remodeling, reduced myocardial infarct size, and ameliorated heart function, have been observed after long-term treatment by functional cardiac patches[6].

Another protective effect of NO has also been attributed to the activation of ATP-sensitive potassium channels ($K_{ATP}$ channels). NO–cGMP-dependent signaling leads to the opening of mitochondrial $K_{ATP}$ channels and reduced calcium overload[43]. The activity of mitochondrial respiratory chain complex II was also evaluated, since it has been proposed as an important regulator of mKATP activity. The reduced complex II activity after nitrate-functionalized patch treatment in this study is consistent with the trend reported by another group[53].

Finally, inflammatory modulation is another function of NO. Inflammation is complicated in MI, leading to myocardial injury and severe fibrosis. In this study, NO delivery by nitrate-functionalized cardiac patches modulated macrophage polarization into the M2 phenotype, which can promote wound healing and tissue repair. Furthermore, NO delivery downregulated the expression of pro-inflammatory cytokines while upregulating pro-repair cytokines involving IL-10, which is an important mediator of tissue repair.

Tolerance represents a major limitation of some organic nitrates utilized in the clinic, especially glyceryl trinitrate (GTN). GTN is the most commonly utilized treatment for various ischemic and congestive cardiac diseases[54–56], although there are also organic nitrates (such as pentaerythritol tetranitrate; PETN) that are devoid of the effects of tolerance emergence[57,58]. Nitrate tolerance is a complex phenomenon and yet to be clearly elucidated. In general, the typical consensus is that nitrate tolerance is mainly caused by oxidative inactivation of the reductase activity of aldehyde dehydrogenase-2 (ALDH2), which results in decreased GTN bioactivation, and the resultant diminished levels of nitrite ($NO_2^-$) and nitric oxide (NO). Recently, Mochly-Rosen et al. have also reported that activators of ALDH2 such as Alda-1 may help to inhibit cardiac injury induced by GTN tolerance[59,60].

In contrast, as a macromolecular nitrate, PCL-ONO$_2$ demonstrates a specific biotransformation pathway that is different from that of small-molecule drugs. First, polymers undergo non-enzymatic hydrolysis to release nitrate anion locally in infarcted myocardium. Then, the generated nitrate anion is reduced to release NO via the nitrate–nitrite–NO sequential pathway. Since the phenomenon of tolerance is not exhibited with the consumption of inorganic nitrate/nitrite[61], it is reasonable to speculate that PCL-ONO$_2$ developed in this study is free from tolerance because the process of biotransformation is different

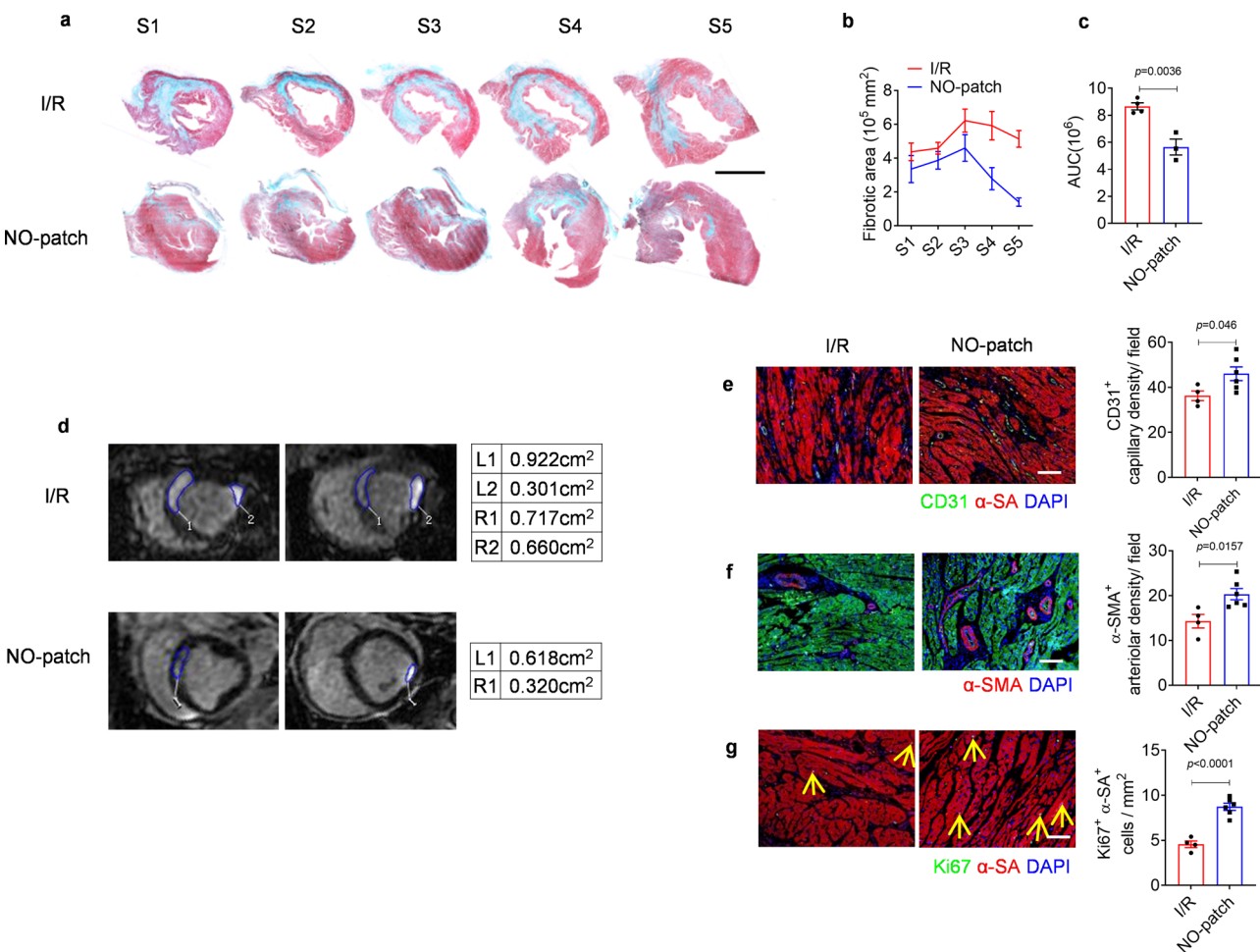

**Fig. 9 Local NO delivery by nitrate-functionalized cardiac patches reduces infarcted size and alleviates adverse ventricular remodeling in pigs. a–c** Masson trichrome staining was performed in sections acquired from five planes from cardiac apex to base (S1-S5, scale bar, 10 mm), and fibrotic area in each section was measured accordingly (**b**), the area under curve (AUC) was calculated to reflect the infarcted size (**c**) ($n = 4$ pigs in I/R group, $n = 3$ pigs in NO-patch group). **d** Representative LGE-cMRI images demonstrating infarcted size 7 days post-surgery, and the infarcted zone was circled and quantified. **e, f** Neovascularization was detected by immunostaining with CD31 and α-SMA, and correspondingly the quantity of capillaries and arterioles was determined. **g** Cardiomyocyte proliferation was detected by co-staining with Ki67 and cardiomyocyte marker α-SA, and double positive proliferating cardiomyocytes were quantified ($n = 4$ pigs in I/R group, $n = 6$ pigs in NO-patch group). Scale bar = 100 μm. Data are expressed as mean ± SEM. Significant differences were detected by two-tailed student's $t$ test. I/R: ischemia/reperfusion; AUC: area under curve.

from that of classic organic nitrates. In addition, nitrate-functionalized PCL does not require additional activators (such as Alda-1), although relevant experiments are required to validate this hypothesis.

Collectively, the stepwise biotransformation of nitrate-functionalized polymers proceeds slowly under physiologic conditions and could be accelerated by microenvironment of the ischemic heart. It enables the local generation of inorganic nitrate in a sustained manner and ischemia-enhanced NO transformation that not only guarantees the conversion efficiency but also abolishes disadvantages associated with the systemic nitrate administration as mentioned before. Besides, the nitrate-functionalized polymer and the cardiac patch have a vast improvement and advantage over organic nitrates administrated by transdermal patch or other routes wherein small molecular drugs rapidly release into the vasculature and the bioconversion to NO mainly occurs through intravascular metabolism[62]. The short half-life of NO (~6 s) restricts its entry into the tissue/organs, thus the therapeutics is mainly based on hemodynamic effects of NO[8]. Compared to organic nitrates, local generation of NO from nitrate-functionalized patches remarkably abolishes

adverse effects, including serious hypotension caused by the vasodilation of systemic vasculature when higher doses of organic nitrates are administrated via oral or transdermal pathways. Local NO delivery can further provide non-hemodynamic effects on the ischemic myocardium, including cardioprotection and immuno-modulation. As a result, different NO conversion pathways lead to the divergent releasing profiles that bring on different therapeutic efficacy finally (Fig. 10).

Furthermore, a head-to-head comparison study has been performed in order to demonstrate the advantages and disadvantages of nitrate-functionalized patches through systemic comparison with both organic nitrate, isosorbide mononitrate (iso-mono) and inorganic sodium nitrate (NaNO$_3$) that have been reported previously by utilizing the rat cardiac ischemia/reperfusion(I/R) model. Ten days after treatment, cardiac histology and function were analyzed. Results show that after intake of sodium nitrate there was a reduction in fibrotic area and infarct size (Supplementary Fig. 18a–c). However, it did not show a positive effect on the cardiac function (Supplementary Fig. 18d, e). In contrast, treatment with isosorbide mononitrate and NO-patch significantly improved the outcomes of cardiac I/R injury,

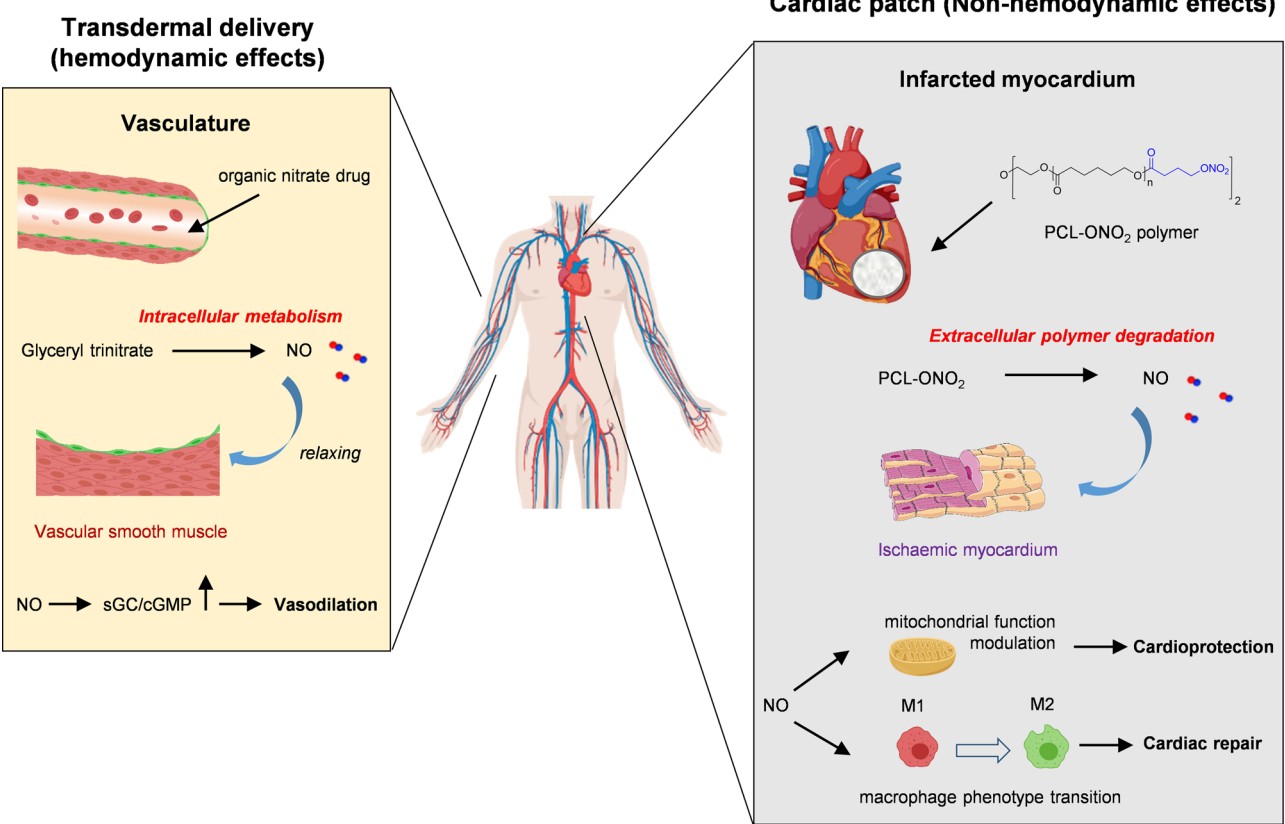

**Fig. 10 Comparison of nitrate-functionalized cardiac patch with organic nitrates delivered by transdermal patch in terms of NO transformation pathway and therapeutic mechanism. Left:** Organic nitrates administrated by transdermal patch or other routes often release into the vasculature rapidly and transform to NO through intravascular metabolism. Due to the short half-life of NO the therapeutics is mainly based on hemodynamic effects through dilation of vascular smooth muscle. **Right:** Compared to small molecular organic nitrates, local NO delivery by nitrate-functionalized patches can further provide non-hemodynamic effects on the ischemic myocardium, including cardioprotection and immunomodulation.

the fibrotic area as well as the infarct size were reduced (Supplementary Fig. 18a–c), and the cardiac ejection function was improved (Supplementary Fig. 18d, e). However, serious cardiac arrest was observed in rats (3/7) after gavage of isosorbide mononitrate which could be caused by hypovolemic shock. As intake of isosorbide mononitrate resulted in whole-body vasodilation and hypotension, which turns to be the risk factor of hypovolemia. Besides, the amount of administrated iso-mono (0.15 mmol) is higher than that of PCL-ONO$_2$ present in the NO patch (0.25 μmol). As a result, despite the comparable therapeutic efficacy, we believe that NO-patch implantation is a promising strategy for cardioprotection and cardiac repair in terms of biotransformation efficiency and safety due to the local NO generation that is responsive to ischemia microenvironment after MI.

The application of our patch will be primarily focused on usage during coronary artery bypass grafting surgery (CABG), wherein the implanted patch played a significant role in repairing the damaged myocardium after revascularization as a consequence of ischemia and ischemia-reperfusion injury (IRI). NO-releasing coatings have been employed in the development of drug-eluting stents[9,10] for coronary heart diseases, and that some of these have entered into pre-clinical trials. However, based on the limited data that has been reported up until now, the therapeutic efficacy of NO-releasing stents is far from satisfactory because of intimal hyperplasia representing the major adverse event following the complete release of loaded NO-drug in longer-term studies[63]. In light of this report, it was long-term NO delivery that proved to be a main challenge. This is because existing drug-eluting systems

have a finite depot of drug and will lose therapeutic function once the depot in depleted[64]. Instead, surface modification of bare stents by immobilizing relevant enzymes to strengthen endogenous NO generation[65] or catalyze the bioactivation of endogenous NO donors[10], such as S-nitrosothiol, may represent a new paradigm for the future of NO-releasing stents.

The relationship between NO and cardiomyocyte proliferation has not been well explored in this study. We indeed detected increased numbers of Ki67$^+$, PH3$^+$, Aurora B$^+$ cardiomyocytes after NO-patch treatment, however, there is no actual division as shown by the formation of midbody observed in Aurora B staining, therefore further studies on the relationship of NO with cardiomyocyte mitosis and cardiac regeneration are needed to reach a consensus conclusion.

In summary, a nitrate-functionalized cardiac patch has been successfully fabricated. The functional patch demonstrated site-specific delivery of NO into the infarcted myocardium under ischemic conditions. In a rat model of MI, administration of NO patches demonstrated optimized therapeutic efficacy, including reduced cardiac injury at an early stage, suppressed adverse cardiac remodeling and ameliorated heart function after long-term treatment, confirming the protective role provided by locally released NO. The translational potential of nitrate-functionalized patches was further evaluated in a porcine ischemia/reperfusion MI model. Cardiac magnetic resource imaging (cMRI) measurements and echography detection reflected remarkably enhanced cardiac function after NO patch implantation, while reduced infarct size and attenuated adverse remodeling were also observed in NO patch-treated hearts.

## Methods

**Synthesis of PCL-ONO$_2$.** All chemicals and reagents were purchased from Sigma-Aldrich (China-mainland), Energy Chemical (China-mainland), and Alfa Aesar (China-mainland), and used directly without further purification. Silica gel (200–300 mesh, Qingdao) was used for column chromatography. $^1$H NMR and $^{13}$C NMR spectra were recorded on a Bruker Avance-400 FT nuclear magnetic resonance spectrometer. Chemical shifts were reported relative to the reference chemical shift of the NMR solvent. The following splitting abbreviations were used: s = singlet, d = doublet, dd = doublet doublet, t = triplet, m = multiplet.

PCL-ONO$_2$ was first synthesized via two steps starting from PCL-diol ($M_n$ = 2000). Briefly, the solution of PCL-diol (5 g, 2.5 mmol) in dry DCM (50 mL) at 0 °C was added sequentially dry TEA (2.07 mL, 15.0 mmol) and 4-bromobutanoyl chloride (1.855 g, 10.0 mmol). The resulting mixture was gradually warmed to room temperature and continued to be stirred for 24 h under argon atmosphere. After completion, the solvent was removed under reduced pressure to give the crude product which was purified by silica column chromatography, eluting with DCM/MeOH (100:4), to give intermediate PCL-Br (3 g) in 60 % yield. $^1$H NMR (CDCl$_3$, 400 M Hz): 3.45 (t, 4H, -CH$_2$-Br), 2.50 (t, 4H, -CO-CH$_2$-), 2.17 (m, 4H, -CH$_2$-CH$_2$-CH$_2$-), 4.06 (t, 35H), 2.31 (t, 35H), 1.65 (m, 70H), 1.38 (m, 35H), the characteristic signals at 4.06, 2.31, 1.65, and 1.38 ppm corresponded to -CH$_2$-protons of PCL backbone, whereas the new peaks at 3.45, 2.50 and 2.17 ppm were assigned respectively to the protons of -CH$_2$-Br, -CO-CH$_2$- and -CH$_2$- in the 4-bromobutanoyl ester unit. $^{13}$C NMR (CDCl$_3$, 101 MHz): 173.54, 64.14, 34.11, 32.71, 28.34, 27.77, 25.52, 24.57, the characteristic signals at 173.48, 64.09, 34.08, 28.31, 25.49 and 24.53 ppm were from PCL backbone, whereas the new peaks at 32.71, 27.77 ppm were assigned respectively to -CH$_2$-Br, -CH$_2$-CH$_2$-Br in the 4-bromobutanoyl ester unit (Supplementary Fig. 2). To a stirred solution of PCL-Br (2 g, 1.0 mmol) in anhydrous acetonitrile (20 mL) was added AgNO$_3$ (1.5 g, 10.0 mmol) at 0 °C. Then the mixture was stirred at room temperature for 24 h. AgNO$_3$ was removed by organic filter and the organic phase was washed with water for three times, dried over Na$_2$SO$_4$ and evaporated under reduced pressure to give the desirable compound as pale waxy solid (1.7 g) with a yield of 85%. $^1$H NMR (CDCl$_3$, 400 M Hz): 4.52 (t, 4H, -CH$_2$-ONO$_2$), 2.50 (t, 4H, -CO-CH$_2$-), 2.17 (m, 4H, -CH$_2$-CH$_2$-CH$_2$-), 4.06 (t, 35H), 2.31 (t, 35H), 1.65 (m, 70H), 1.38 (m, 35H), the new peaks at 4.52 was assigned to the protons of -CH$_2$-ONO$_2$ in the 4-(nitrooxy)butanoyl ester unit. $^{13}$C NMR (CDCl$_3$, 101 MHz): 173.48, 72.05, 64.09, 34.07, 28.30, 27.75, 25.49, 24.53, the characteristic signals at 173.48, 64.09, 34.07, 28.30, 25.49, and 24.53 ppm were from PCL backbone, whereas the new peaks at 72.05, 27.75 ppm were assigned respectively to -CH$_2$-ONO$_2$, -CH$_2$-CH$_2$-ONO$_2$ in the 4-(nitrooxy)butanoyl ester unit. (Supplementary Fig. 3).

**Synthesis of NO probe (DAC-S).** NO sensitive probe synthesized according to a reported method[30]. Briefly, 4-Amino-3-nitrophenol (100 mg, 0.65 mmol) and NaH (60% in mineral oil) (30 mg, 0.65 mmol) were dissolved in anhydrous DMF (20 mL). The mixture was stirred at room temperature for 20 min under an argon atmosphere. Then a solution of IR-783 (200 mg, 0.27 mmol) in anhydrous DMF (10 mL) was added to the reaction via a syringe. The reaction mixture was further stirred for 6 h. The mixture was extracted with ethyl acetate twice (30 mL x 2), and the combined organic phase was washed with water (20 mL), brine (20 mL) and dried over Na$_2$SO$_4$. After removal of the solvent under reduced pressure, the crude product was purified by silica gel chromatography with 30% CH$_3$OH in DCM to afford the intermediate (IR-NO$_2$) as a dark green solid (180 mg, 80%).$^1$H NMR (400 MHz, CD$_3$OD): δ 7.89 (d, 2H, $J$ = 14.3 Hz), 7.61 (d, 1H, $J$ = 2.9 Hz), 7.08-7.31 (m, 9H), 6.97 (d, 1H, $J$ = 9.3 Hz), 6.12 (d, 2H, $J$ = 14.3 Hz), 4.04 (t, 4H, $J$ = 6.3 Hz), 2.78 (t, 4H, $J$ = 7.0 Hz), 2.67 (t, 4H, $J$ = 5.9 Hz), 1.80-1.96 (m, 10H), 1.33 (s, 12H). $^{13}$C NMR (101 MHz, MeOD): δ 172.21, 163.43, 149.77, 142.20, 141.66, 141.06, 130.14, 128.43, 124.80, 124.61, 122.15, 121.96, 121.09, 110.77, 108.90, 99.98, 50.40, 48.91, 43.54, 35.57, 26.83, 25.82, 23.89, 22.22, 21.08. MS (FAB$^+$): 845 (M – Na$^+$ + 2H$^+$) (Supplementary Fig. 8). To a solution of IR-NO$_2$ (100 mg, 0.12 mmol) in MeOH (4 mL) and concentrated HCl (0.6 mL) was added SnCl$_2$·H$_2$O (450 mg, 2.0 mmol). The solution was stirred at room temperature overnight under an argon atmosphere, then neutralized with 2 N NaOH. The precipitate was removed by filtration and the filtrate was concentrated under reduced pressure and the crude product was purified by silica gel chromatography with 35% CH$_3$OH in DCM to afford the desired product as a dark green solid (40 mg, 40%).$^1$H NMR (400 MHz, CD$_3$OD): δ 8.07 (d, 2H, $J$ = 14.2 Hz), 7.06-7.28 (m, 8H), 6.68 (d, 1H, $J$ = 8.4 Hz), 6.55 (d, 1H, $J$ = 2.9 Hz), 6.31 (dd, 1H, $J$ = 8.4, 2.9 Hz), 6.15 (d, 2H, $J$ = 14.2 Hz), 4.11 (t, 4H, $J$ = 5.5 Hz), 2.87 (t, 4H, $J$ = 6.7 Hz), 2.73 (t, 4H, $J$ = 5.8 Hz), 1.82-2.14 (m, 10H), 1.41 (s, 12H), $^{13}$C NMR (101 MHz, MeOD): δ 172.22, 172.02, 171.75, 142.54, 142.39, 142.25, 141.11, 140.89, 128.47, 124.57, 122.65, 122.55, 122.01, 110.54, 99.22, 63.98, 50.49, 48.88, 43.42, 27.34, 26.89, 25.84, 23.94, 22.27, 21.11. MS (FAB$^+$): 815 (M – Na$^+$ + 2H$^+$) (Supplementary Fig. 9).

**Fabrication and characterization of nitrate-functionalized cardiac patch.** Briefly, high molecular weight PCL (PCL80K) was mixed with PCL-ONO$_2$ (PCL2K) at blending ratios of 9/1 (w/w). The mixture was dissolved in mixed chloroform/methanol (5:1, v/v) by sufficient stirring to obtain homogeneous solution with final concentrations of 10 wt% (w/v). The electrospun mats were fabricated under the following processing conditions: needle tip–collector distance = 15 cm, solution flow rate = 2 mL/h, and voltage = 15 kV. The as-prepared mats were dried in vacuum at room temperature for more than 3 days in order to remove the residual solvents sufficiently.

The surface morphology of electrospun mats was observed under a field emission scanning electron microscopy (SEM; Quanta 200, Czech) at an accelerating voltage of 10 KV. The surface was sputter-coated with gold before observation.

The surface water contact angle of electrospun cardiac patch was measured by the sessile drop method using a Harke-SPCA goniometer (Beijing,China). Samples were adhered on glass slice by double-face adhesive tape and put onto the sample stage. The images were recorded continuously after water dropping on film 10 s at room temperature. The average values were obtained from 5 measurements at different positions.

Mechanical tensile testing was performed on an Instron universal tensile tester (model 5865). The electrospun mats were cut into specimens in rectangular shape with dimension of 15 × 10 × (~0.6) mm. The distance between two grips was set as 10 mm. A tensile test was performed at ambient temperature with crosshead speed of 10 mm/min. Each test was repeated on 3 specimens.

**In vitro degradation.** Briefly, 10 mg/mL nitrate-functionalized materials were incubated in PBS buffer at 37 °C, then the production of nitrate and nitrite anions was evaluated according to the Griess methods using Total Nitric Oxide Assay Kit (Beyotime Biotechnology, S0023). The optical density was measured at 540 nm, and the amount of nitrite was determined using sodium nitrite as a reference standard.

**In vitro NO release detection.** Briefly, 10 mg/mL nitrate-functionalized PCL and 5 µM nitric oxide fluorescence probe DAF-FM DA were co-incubated in heart tissue homogenate (10 mg/mL) at 37 °C. Then, the fluorescent intensity of the reaction mixture was measured at different time points. Data were acquired through microplate reader with, $\lambda_{ex}$ = 490 nm and $\lambda_{em}$ = 520 nm.

**Subcutaneous NO release detection.** Circular samples (6 mm in diameter) were implanted subcutaneously into the dorsal skin of the rat after recording the weight, and at pre-determined time points, the implants were recollected. Then the surrounding tissue was peeled off, followed by washing with sterile saline and air dried. The residual NOx content was assessed by using chemiluminescence NO analyzer (NOA) (Seivers 280i, Boulder, CO). In brief, the samples were immersed in 5 mL of Vanadium (III) chloride (50 mM/L). The generated NO gas was diffused in the test solution and transported to NO analyzer by a stream of N$_2$ (g). The calculation of the generated NO was based on the calibration curves of the NOA, which was described in detail elsewhere[66].

**Rat MI model.** Male Sprague-Dawley (SD) rats (8-week old) were used according to the Animal Use Guidelines. All procedures for rat experiments were approved by the Animal Care and Use Committee of Nankai University. To ensure comparable severity of injury and injury size between groups, rats without pre-determined assignment received LAD ligation at a position 2 mm below the cross-junction of the left atrial appendage and the pulmonary conus, and patch implantation was performed randomly by a surgeon blind to the groups. The rat model of acute MI was established as described previously. Rats were anesthetized via intraperitoneal injection of 10% chloral hydrate (350 mg/kg), followed by fixation to a heating pad (37 °C) at supine position. Endotracheal intubation was performed and rats were ventilated with a mechanical respirator (Hallowell EMC Microvent I) setting at 110 breaths per minute with a tidal volume of 6 ml. After removing the fur covering the chest, an oblique incision of skin from xiphoid towards mid-axilla was made. The muscles were separated by blunt operation without cutting to expose thorax. Then the heart was exposed through a thoracotomy between the third intercostal and the LAD was ligated with 6-0 suture. Cardiac ischemia was confirmed by the observation of pallor and cyanosis in left ventricular. Immediately after LAD ligation, cardiac patches (4 mm × 6 mm) were fixed closely to cover the infarcted myocardium with 8-0 suture. Then the incision was closed and the animals were allowed to recover from anesthesia. Rats in MI group only received LAD ligation without patch implantation, while sham operated rats only experienced thoracotomy without LAD ligation and patch implantation. Buprenex (0.01 mg/kg) and Carpofen (5.00 mg/kg) were administrated to relief pain and distress after procedures.

To create a chronic injury model, MI was induced in rats, and 4 weeks later, echo was measured and followed by the secondary open chest surgery to implant patches.

**Rat cardiac ischemia/reperfusion model.** Rat model of cardiac ischemia/reperfusion was induced by ligation of left ascending artery for 30 min, followed by removal of the suture. After closing the incision, the rat was allowed to recover. Then the isosorbide mononitrate (75 mg/kg/day)[67] or sodium nitrate (100 mg/kg/day) dissolved in sterilized saline was medicated through gavage, twice a day at an interval of 12 h. For the patch group, the patch was implanted immediately after reperfusion.

**In vivo NO release detection**. Electron paramagnetic resonance (EPR) assay was employed to measure the levels of NO in heart tissues as previously[26]. Briefly, one day after surgery, the rats were anesthetized by I.P. injection of 10% chloral hydrate (350 mg/kg). DETC sodium salt (500 mg/kg, Sigma-Aldrich) was then administrated by intraperitoneal injection in distilled deionized water (250 mM). Five minutes later, ammonium ferrous sulfate (50 mM) citrate solution (250 mM) was injected subcutaneously (2 ml/kg). After 1 h, the hearts were collected and immediately dissected into infarcted part and distal part, which were frozen separately in liquid nitrogen. Subsequently, the frozen heart tissues were crumbled into small pieces in homogenizer tubes and extracted with ethyl acetate (200 μL) immediately. The ethyl acetate extract was concentrated and transferred to 50 μL capillary tube and measured on x-band EPR at room temperature (23 ± 1 °C). The following acquisition parameters were used: modulation frequency, 100 KHz; microwave power, 10 mW; modulation amplitude, 2 G; number of scans, 30. The double integrated area of EPR spectrum was calibrated into concentrations of NO-Fe $(DETC)_2$ using TEMPO as a standard. EPR spectral simulation was conducted by the WINSIM program.

**Ex vivo NIR imaging**. NO sensitive probe (DAC-S) was dissolved in sterile saline to make a working solution of 10 μM. One day after surgery, the rats were anesthetized again. After intubation and thoracotomy, the heart was exposed, and DAC-S probe was injected into myocardium at three sites involving the infarcted core and both border zones. The injection volume is 10 μL in each site. After 1 h, the heart was collected and cut into pieces for ex vivo imaging. NO specific signal was detected by using CRI Maestro noninvasive fluorescence imaging system. All images were acquired through 780–950 nm range in 10 nm steps using the "ICG" filter with excitation at 735 nm (exposure time 1000 ms; $n = 3$ for each group).

**Cardiac function measurement**. Transthoracic echocardiograph was performed by using the Vevo 2100 Imaging System equipped with a 20 MHz transducer (FujiFilm VisualSonics, Inc. Toronto, Canada) by researchers not knowing the groups as previously described[68]. The base line of cardiac function of rats was recorded three days before surgery. When measuring the cardiac function, all animals were anesthetized by isoflurane inhalation mixed with oxygen, and M-mode images were acquired. Then the left ventricular internal diameter at end-diastole (LVIDd) and systole (LVIDs) were measured, and accordingly, the LV-ejection fraction (LV-EF), LV-fractional shortening (LV-FS), LV end-diastole volume (LV-EDV), and LV end-systole volume (LV-ESV) were calculated as indicators of LV function and structure. All measurements were recorded from 5 continued cardiac cycles.

**Measurement of myocardial blood flow**. Myocardial blood flow was measured by using method of fluorescent microspheres perfusion[69]. Briefly, 500000 microspheres of 15μm in diameter that labeled with red fluorescence (Molecular Probes, Invitrogen, CA, USA) were injected into left ventricular over more than 20 s. Forty-five minutes later, the hearts were removed and fixed in 4% paraformaldehyde over 24 h. For the histological determination of myocardial blood flow, fixed hearts were immersed in 30% glucose for more than 48 h and subsequently embedded with OCT compound. Frozen sections in 20 μm thickness were made. The number of fluorescent microspheres in the ischemic border zone was counted under a fluorescent microscope to indicate the relative blood perfusion volume.

**Mitochondrial complex activity measurement**. Enzyme activities of mitochondrial respiration chain complex I and complex II were measured according to the operation manual (Solarbio, Beijing, China). Heart tissues harvested at day 1 were first made into 10% homogenate, and the mitochondrion was isolated after centrifuge and sonication. After mixing with its corresponding substrate solution, the activities of complex I and complex II were detected spectrophotometrically, respectively.

The tissue level of $H_2O_2$ was measured with 10% homogenates by using Hydrogen Peroxide/Peroxidase assay kit (Nanjing Jiancheng Bioengineering Institute, China).

**Detection of NO generation by macrophages**. For imaging the endogenous NO in RAW264.7 murine macrophages, the cells were pre-treated with 100 ng/mL LPS for 12 h in DMEM medium at 37 °C, and then incubated with 5 μM NO fluorescence probe DAF-FM DA (Solarbio, Beijing, China) for 2 h at 37 °C in the same medium. Images were acquired through Olympus Fluo View™ FV1200 confocal microscope with band path of 500–600 nm upon excitation at 488 nm.

**Measurement of pO2**. To measure cardiac $pO_2$ levels, rats were ventilated and anesthetized with inhalation of isoflurane. A heating pad was used to keep the body temperature to be 37 °C. After expose of the heart, OxyLite probes were punctured into the myocardium and readouts were taken for one minute after stabilization. The mean values were used to indicate the level of $pO_2$.

**Measurement of pH**. Cardiac pH levels was analyzed with $^{31}P$ magnetic resonance spectroscopy, one day after surgery, the rats were anesthetized by I.P. injection of

10% chloral hydrate (350 mg/kg). The hearts were collected and immediately dissected into infarcted part, prepare into homogenate with $D_2O$ for detection.

**Measurement of S-nitrosylated proteins**. One day after patch implantation, heart sections were used for detection of S-nitrosylation by using commercially available kit (S-Nitrosylated protein detection kit, Cayman, 10006518). To proceed with this assay, heart sections were firstly incubated with blocking reagent to block free thiols in the samples, and then followed by reduction and labeling with maleimide-biotin as well as fluorescein-avidin, S-nitrosylated proteins can be localized.

**Histological analysis**. Histological analysis were performed by researchers blind to the groups. At indicated time points, rats were anesthetized via intraperitoneal injection of 10% chloral hydrate (350 mg/kg) following sedation with inhalation of isoflurane and fixed in surgery plate in supine position. Hearts were fixed through trans-cardiac perfusion with saline followed by 4 % paraformaldehyde (PFA), subsequently, the heart samples were immersed into 4% PFA and fixed over 24 h. Latterly, the heart samples were embedded into paraffin blocks and cut into 5 μm thickness sections. For frozen sections, heart samples need to be immersed in 30 % glucose over 48 hours after fixation with PFA, and then embedded with OCT. The frozen sections were also 5 μm in thickness, and stored at −20 °C. Hematoxylin-Eosin (H&E) staining, Masson trichrome staining, Nagar-Olsen staining and Sirus Red staining were performed with paraffin sections following a standard protocol. The cell apoptosis was detected by using DeadEnd™ Fluorometric TUNEL System (Promega, G3250) according to the operating manual, after which the sections were incubated with cardiomyocyte marker α-sarcomeric actin (α-SA) (Abcam, ab9465). DAPI was used for nucleus staining. For immunofluorescence staining, frozen sections were washed three times with PBS, and then blocked with goat serum at room temperature for 1 h. Antibodies against α-SA(Abcam, ab9465, 1:100), CD31 (Abcam, ab28364, 1:100), α-SMA (Boster, BM0002, 1:500), CD68 (Abcam, ab201340, 1:100), CD86 (Abcam, ab234000, 1:100), CD206 (Abcam, ab64693, 1:100), Ki67 (Abcam, ab16667, 1:200) diluted in goat serum were then dropped to cover sections and incubated overnight at 4 °C. Then the sections were washed three times with PBS, followed by incubating with Alexa Fluor 594-conjugated goat anti-mouse IgG (Abcam, ab150116, 1:200) and Alexa Fluor 488-conjugated goat anti-rabbit IgG (Abcam, ab150077, 1:200) for 2 h at room temperature. After washing with PBS, DAPI-containing fluoromount-G (Southern Biotech, 0100-20) was used to mounting.

**qRT-PCR**. The rats were sacrificed three days after surgery through $CO_2$ inhalation, and the hearts were collected. The muscle tissues of left ventricular in the border zone were dissected and frozen in liquid nitrogen, after which the total RNA was extracted using Trizol reagent (Invitrogen, Grand Island, NY). RNA yield was determined by using NanoDrop spectrophometer (NanoDrop Technologies). First-strand cDNA was synthesized using ologo dT primers with reverse transcriptase (TransGen Biotech, China). Subsequently, real-time qPCR was performed on a CFX96 Real-Time PCR System (Bio-Rad, Hercules, CA). The relative expression of mRNA of interest was expressed as $2^{-(\triangle\triangle CT)}$ and normalized to housekeeping gene GAPDH. Each independent experiment was performed in triplicate. The primer sequences used in this study are listed in Supplementary Table 2.

**Porcine MI model**. All procedures were approved by the Ethics Committee of Shanghai Jiaotong University Animal Department. Pigs weighing 7–8 kg were used in this study. Briefly, intramuscular injection of Zoletil was performed to induce anesthesia. After orotracheal intubation, 2% isoflurane was supplied to maintain anesthesia during the surgery. Electrocardiogram (ECG), arterial oxygen saturation, blood pressure, and heart rate were monitored throughout the operation. After a thoracotomy in the left chest, the heart was exposed and ischemia was introduced by circling the LAD before the branch of second diagonal coronary arteries, and the suture was loosed to achieve reperfusion after 90 min ischemia. Cardiac patch (20 mm × 15 mm) was implanted immediately after ischemia in the experimental group, while I/R injured pigs without patch implantation were used as the control group.

To confirm the cardiac injury, at the preliminary models, blood samples were drawn before and after surgery, and plasma cTnI and CK-MB levels were measured.

**Echocardiography**. To detect the cardiac function, transthoracic echocardiography (ECG) was performed before and after surgery by using an echocardiographic system (Phillips CX50) equipped with a S5-1 sector array transducer as previously described. The short-axis M-mode images were presented and dimensions of left ventricle at both diastole (LVIDd) and systole (LVIDs) were measured, and accordingly, the ejection fraction (EF), fraction shortening (FS) and end-systole volume (ESV) were calculated.

**3T-cMRI acquisition**. MRI measurement was performed by operators not knowing the assignment of the groups. Anesthesia was achieved by applying an intramuscular injection of a mix of tiletamine hydrochloride and zolazepam

hydrochloride (Zoletil®50, Virbac S.A., France). The studies were performed on a 3.0T-CMR system (MR750, GE Healthcare, Waukesha) by operators blinded to the study medication. Animals were positioned in a head-first supine position with a flexible phased-array surface coil placed over the chest. ECG gating was used to acquire still images of the heart. The following dedicated CMR sequences were acquired in all cases: "cine" (b-SSFP) imaging sequence to assess wall motion and cardiac function, late gadolinium enhancement to assess the extent of myocardial necrosis. All cMRI assays followed the same sequences. First, scout images (T1-TFE sequence) were obtained to localize the true axes of the heart and define a field of view involving the whole heart. Afterward, the b-SSFP cine imaging was performed in both horizontal and vertical long axes (4- chamber and 2-chamber views) and in multiple contiguous short-axis images covering the whole LV. In the short-axis cine sequence 30 cardiac phases of every slice were acquired to guarantee a correct evaluation of the wall motion and heart function. Thereafter, a gadolinium-based contrast agent was injected intravenously (Gd-GTPA, Magnevist®, Berlex Laboratories Inc., Wayne, New Jersey, USA) at a dose of 0.1 mmol/Kg. The early gadolinium enhancement sequence was acquired 1 min after the administration of the contrast. The late gadolinium enhancement (LGE) sequences were obtained 10 min after administration of the contrast.

All CMR images were analyzed using dedicated software (Circle CVI42 Calgary) by a CMR-trained radiologist blinded to the study medication. Briefly, LV cardiac borders were traced in each image of the cardiac phases representing the end diastole and end systole to obtain the left ventricle end-diastolic- and end-systolic volumes (LVEDV and LVESV, respectively) and LVEF. Infarct size (necrosis) was quantified from the extent of myocardial enhancement in the LGE CMR sequence (see below for the detailed parameters).

**Histological analysis**. Four weeks after surgery, pigs were anesthetized by inhalation of isoflurane, and pentobarbital solution (390 mg/mL, 1 mL/5 kg) was injected to euthanasia. Heart tissues were harvested for histological analysis. To measure infarct size, pig hearts were cut into five evenly separated slices from cardiac apex to base, and each slice was used for paraffin section. Masson staining was performed with all the sections, while immunostaining was performed with the middle one. Masson staining and immunostaining were performed according to a standard protocol in our lab. Briefly, paraffin sections were dewaxed and rehydrated, followed by hematoxylin staining and acid ponceau staining. After differentiation with 1% phosphomolybdic acid solution, brilliant green staining was introduced. Then the sections were dehydrated and mounted with neutral balsam. In immunostaining, antigen retrieval was performed by boiling sections in citrate buffer (0.01 M, pH 6.0), followed by washing and blocking, the primary antibodies was incubated overnight at 4 °C. Then the corresponding fluorescein-conjugated secondary antibodies were dropped and incubated for 2 h at room temperature. Images were acquired under Zeiss Axio Imager M2 Advanced Microscope Platform. More than ten images acquired in each animal were used for quantification.

**Date acquisition and statistics**. Data acquisition were performed by investigators who are blind to the groups, and the related data sets were acquired and proceeded with Image J version 1.8.0, Zeiss Axio Imager M2 Advanced Microscope Platform, CRI Maestro noninvasive fluorescence imaging system, Circle CVI42 Calgary software, OxyLite monitoring System, Excel 2016(Microsoft), Vevo 2100 Imaging System(FujiFilm VisualSonics), Image Pro Plus 6.0. All statistical analyses were performed through commercial software GraphPad Prism 7, and data are presented as mean ± SD or mean ± SEM. Normal distribution was first checked, and comparisons between two groups were performed by Student's t test, while for multiple group comparison, one-way or two-way ANOVA analyses was performed. For all the statistical analyses, significance was accepted at $p < 0.05$.

**Reporting summary**. Further information on research design is available in the Nature Research Reporting Summary linked to this article.

## Data availability
The data supporting the findings from this study are available within the manuscript and the supplementary information. Any remaining raw data are available from the corresponding author upon reasonable request. Source data are provided with this paper.

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

## Acknowledgements
This study is supported by the National Key R&D Program of China (2018YFE0200503), National Natural Science Foundation of China (Nos. 81925021, 81921004, 81873923, 81973269, and 81871500), and Science & Technology Project of Tianjin of China (No. 18JCJQJC46900).

## Author contributions
J.S. and Q.Z. conceived the original concept and initiated this project. Q.Z. designed the experiment and supervised the entire project. J.H. and M.Q. carried out the synthesis of all compounds and fluorescence probes. H. W. prepared the cardiac patch under the supervision of Q.Z. F.W. and M.Q. performed vascular physiological function assay. Y.P. and M.Y. established porcine MI model. D.Z., D.J., T.H., and S.W. performed the in vivo evaluation and analyzed data under the supervision of Q.Z. and M.Y. S. L. carried out RT-PCR assay. L.W. and Y.Z. performed ultrasound and MRI analyses, respectively. D.K., Y.C., and Z.Y. helped in data collection. J.H. provided critical feedback and helped in the review of the article. Q.Z., D.Z., and M.Q. wrote the paper with input from other authors.

## Competing interests
The authors declare no competing interests.
