## [Peer Review File · Nature Communications]

REVIEWER COMMENTS

Reviewer #1 (Remarks to the Author):

The authors designed a PLA-based NO-releasing patch as a therapy for acute myocardial infarction. They show that the NO patch releases nitrite and nitrate in vitro, release the NO species over a period of >28 days in vivo (dorsal skin model in rats), show that it generates higher NO concentrations in the infarcted than in sham myocardium (rats), that it reduces infarct size and improves function in a permanent LAD ligation model in rats and an I/R model in pigs. The beneficial effects were associated with less inflammation, fibrosis, hypertrophy and a tendency towards better survival in pigs. Overall, the authors suggest the NO patch as a novel therapy of MI.

I find this overall a very interesting and innovative approach. The concept of local delivery of NO species is attractive, particularly the evidence for increased release under hypoxic/ischemic conditions. While the data seem overall convincing, there are aspects of concern.

1. A corner stone of the therapeutic concept is that local ischemia augments delivery of NO species. While the present data are suggestive of this idea, the mechanism has to be shown more directly. What for example is the effect of acidic pH in vitro? What is the pO₂ in the NO patch implanted on non-infarcted sham rats? One would expect hypoxia to occur also under these conditions.
2. The degree of protection is striking and a bit difficult to believe. How do the authors ensure equal injury sizes? Given the small number of animals, even minor differences in the proximity of LAD ligation can largely confound the results.
3. I did not find measures against bias such as blinding and randomisation procedures.
4. It would strengthen the study if key experiments e.g. in rats were reproduced by another group.
5. Fig. 5D: Measurement of CM size by H&E staining is not state-of-the-art, please supplement with wheat germ agglutinin or dystrophin staining which better demarks cell borders.
6. It is difficult to understand how NO should stimulate CM proliferation, which is extremely low in the postnatal heart and not or only very mildly higher in injured hearts (see consensus statement in *Circulation* 2017; PMID:28684531). I am not aware that NO has a convincing effect on CM proliferation (in contrast to interventions into the Hippo pathway). Thus, the marked effect shown in Fig. 5G and 7G is difficult to reconcile. Moreover, the measurement of Ki67 in histological sections is clearly not sufficient to proof CM proliferation. Please provide higher resolution confocal images of PH3 and aurora B kinase stainings and/or FACS staining for a CM marker such as actinin, troponin and Ki67 or PH3. It must also be differentiated between polyploidy and multinucleation and true cytokinesis, probably best by aurora B kinase staining.
7. For a therapy to be clinically applicable, effects of the NO patch would need to be shown in a model of chronic injury, e.g. implantation 4-8 weeks after MI. It is hard to imagine that patches were applicably in the settings of an acute MI.

8. I did not find a COI statement.

Reviewer #2 (Remarks to the Author):

In the work "Nitrate-functionalized patch confers cardioprotection and improves heart repair after myocardial infarction via local nitric oxide delivery" by Zhu et al., the authors have prepared a biodegradable polymer functionalized by nitrate as the NO donor for the treatment of cardiovascular disease. The authors have confirmed the NO release in vivo using fluorescence probes, and checked the effect on cardiac disease in pig. However, many similar works on the biodegradable polymer NO donors have been reported, such as "ACS Macro Lett. 2019, 8, 1552-1558", "Polymer Bulletin 2018, 75, 2971-2985", and "Biomater. Sci., 2013, 1, 625-632". The authors have not demonstrated any advantages over these works. Consequently, this work should be rejected due to the lack of novelty.

Additional comments:

1. The detail characterizations for the polymer (PCL-ONO₂) and NO probe were necessary, therefore the ¹³C NMR and MS data should be provided.

2. The NO-sensitive probe was used to acquire the NIR fluorescence images of infarcted myocardium by the response of the probe to NO. The response of the probe to NO which was released from the in vitro degradation of polymer (PCL-ONO₂), should be evaluated. In addition, why did the author choose three sites instead of more sites at myocardium to perform ex vivo NIR imaging (Fig. 2c)?

3. How to assess the toxicity of PCL-ONO₂ in the treatment ?

4. In page 4 and 5, the authors said "The quantitative data indicated that in the distal region, there was no detectable difference in NO levels among the various groups (Fig. 2c), but in the infarcted myocardium, NO levels were evidently enhanced in the MI group due to endogenous NO production by activated macrophages." This meaning that fig 2d, the high concentration of NO in MI+NO-patch group was produced from the activated macrophages not from the PCL-ONO₂ system. This is contradictory, please give the reasonable explanations.

Reviewer #3 (Remarks to the Author):

Comments

1) General comment: The entire manuscript needs English proof correction (see examples in the abstract below).

2) General comment: I understand the rationale to use these nitrate patches that can be implanted nearby the heart, where infarct damage is most severe. This especially allows to reach higher local doses of the nitrate, which otherwise cannot be given at very high oral doses as this would cause serious hypotension with life-threatening loss of blood pressure. I am not sure, whether this most important benefit of a nitrate-containing biomaterial that can be implanted, was sufficiently addressed in the present manuscript. In the Introduction a respective rationale for using the drug-containing patch, rather than using oral administration of the nitrate was chosen in the present work. In the Discussion, please add the pros and cons of the nitrate-biomaterial versus oral administration.

3) General comment: NO-release from biopolymers was already used in the drug-eluting stents (DES), which play an important role for prevention of MI but also first-line treatment after MI. This important point is not mentioned at all in the present work, which represents a major weakness and also implies that the authors did not read the essential background on NO and nitrates usage against MI damage in the clinics. I think that NO-eluting stents did not prove to be superior to bare metal stents – what would this mean for your proposed concept here?

4) General comment: Also nitrate tolerance, which represents a major limitation of the clinical use of organic nitrates, was not mentioned at all, which represents a major weakness and also implies that the authors did not read the essential background on NO and nitrates usage against MI damage in the clinics. The authors should introduce and discuss this major adverse side effect of nitrate therapy (for review see PMID: 21576678, PMID: 23864131, PMID: 26261901) as it may also represent another strength of the here proposed NO-releasing patch that can be implanted as I assume that implantation of the nitrate-containing patch is not subject to nitrate tolerance, which may represent a major advantage besides the higher local concentrations of NO as compared with oral nitrate administration.

5) General comment: The authors briefly mention organic nitrate biotransformation/bioactivation, which is a crucial step in the pharmacological action (NO release) of a given organic nitrate. The authors should make themselves familiar with the routes of organic nitrate bioactivation (for review see PMID: 21576678, PMID: 23864131, PMID: 26261901). As they correctly mention, the nitrate-containing biomaterial cannot cross the cell membrane (as low molecular weight nitrates do) and reach the place of bioactivation (usually an enzyme such as aldehyde dehydrogenase, P450s or

potentially xanthine oxidoreductase). Instead, the authors propose spontaneous hydrolysis and inorganic nitrate release from the nitrate-containing biomaterial. Then however, this method would compete with oral administration of inorganic nitrate (which can be administered at high doses, e.g. via vegetables such as beetroot or spinach). What is then the advantage over oral inorganic nitrate administration, which also proved beneficial to prevent MI or ameliorate MI damage? This needs to be explained in detail and also the bioactivation of organic and inorganic nitrates should be better explained.

6) General comment: As inorganic nitrate should be the “active” species released from the nitrate-containing biomaterial, the authors should introduce and discuss the current knowledge on inorganic nitrate bioactivation and reported beneficial effects in MI and other heart disease models (for review see PMID: 26265312, PMID: 18167491).

7) General comment: Overload with NO is also not good and termed “nitrosative stress”. Did the authors observe any signs of nitrosative stress in their model (e.g. dramatic increase in S-nitrosated proteins, some SNO modification cause loss of function in enzymes)?

8) General comment: Whereas the presented data are impressive, the authors need to present a head-to-head comparison of protective effects of the nitrate-containing patch versus oral or intravenous organic nitrate administration versus inorganic nitrate treatment during ischemia/reperfusion. How is infarct size and cardiac function affected by these three treatment protocols in the same infarct model? There are also organic nitrates that are devoid of nitrate tolerance and therefore could be used at high dose (e.g. PETN, see PMID: 11527645, PMID: 24071762).

9) General comment: Previously, it was shown that already improvement of organic nitrate bioactivation and prevention of nitrate tolerance may be sufficient to conserve the anti-ischemic effects and protection by organic nitrates such as nitroglycerin in a MI model (PMID: 18787169, PMID: 22049071). How would your nitrate-containing patch compete with this pharmacological activation of nitroglycerin bioactivating enzyme ALDH-2 by Alda-1?

10) General comment: I have seen much more impressive improvements by some treatment protocols in MI animal models in the past than those survival data presented in figure 6. Therefore, it is really important to compare head-to-head different treatment protocols (see my suggestions above, e.g. oral organic nitrate, inorganic nitrate, Alda-1) to demonstrate the advantage of the here presented method over those previously reported.

11) Abstract: „wherein the nitrate pharmacophores were covalently bonded to biodegradable polymers“ – „pharmacophores“ is unclear meaning, better use „pharmacologically-active groups“ or „pharmacological functional groups“. Also „bonded“ sounds odd, better use „bound“.

12) Abstract: „infarcted zone“ should be „infarcted area“ or „infarcted tissue“.

13) Abstract: “with therapeutic mechanism different” – sounds odd, better use “by therapeutic strategies different”.

14) Other specific concerns: The number of animals in many of the measured parameters is very low (n=3), which hardly allows a full statistical analysis.

15) Other specific concerns: why was survival only tested in the pig model with very low amount of animals? I think with the rodents it would have been easier to have large animal numbers for the survival studies.

16) Animals: Please provide the ethical permission number for the different animal experiments.

17) Animals: What kind of analgesia was administered to the animals after MI surgery. As they feel pain afterwards, the international animal protection guidelines recommend to use analgetic drugs after MI (LAD ligation) surgery.

18) Animals: what method was used for sacrifice of the animals and what anesthesia was used for that?

Reviewer #4 (Remarks to the Author):

In this manuscript, the authors developed a nitrate conjugated PCL patch for treating ischemic heart diseases. Nitrate conjugated PCL patch was hydrolyzed under inflammatory micro-environment after myocardial infarction, nitrate ion was released and further converted to nitric oxide. In vivo data demonstrated cardioprotection effects by using NO-patch on rat and porcine myocardial infarction model. However, the authors did not provide compelling rationale for utilizing cardiac patch on therapeutic molecules delivery.

Specific comments:

1. Using cardiac patch can provide scaffolding for culturing therapeutic cells (e.g., cardiac cells) and be used as an implantation to repair injured heart. As for covalently conjugated drug delivery platform, cardiac patch requires more time to release the therapeutic molecules. Compared to nano/microparticles system, what is the main advantage to use cardiac patch to deliver therapeutic molecules?

2. Although the water contact angle slightly decreased after nitrate conjugation, NO-patch was still relatively hydrophobic (water contact angle was $\sim 120^\circ$). It is unclear how this property would be favorable for the tissue microenvironment.

3. The subcutaneous NO release could only last for 120 minutes (figure 1f). It is quite short -lived. How does this translate into therapeutic benefits?

4. The authors described that the elevated NO levels in the infarcted area was due to endogenous NO production via activated macrophages. However, there was no evidence showing the increased NO was produced by macrophages.

5. There was a gap between NO-patch and myocardium in supplementary figure 9. Would this gap prevent leukocytes infiltration into NO-patch?

6. It is not clear what is 5G in figure 2b.

7. It is difficult to compare cardiomyocyte size in figure 3d. More representative images are needed.

Reviewer #1 (Remarks to the Author):

The authors designed a PLA-based NO-releasing patch as a therapy for acute myocardial infarction. They show that the NO patch releases nitrite and nitrate in vitro, release the NO species over a period of >28 days in vivo (dorsal skin model in rats), show that it generates higher NO concentrations in the infarcted than in sham myocardium (rats), that it reduces infarct size and improves function in a permanent LAD ligation model in rats and an I/R model in pigs. The beneficial effects were associated with less inflammation, fibrosis, hypertrophy and a tendency towards better survival in pigs. Overall, the authors suggest the NO patch as a novel therapy of MI. I find this overall a very interesting and innovative approach. The concept of local delivery of NO species is attractive, particularly the evidence for increased release under hypoxic/ischemic conditions. While the data seem overall convincing, there are aspects of concern.

1. A corner stone of the therapeutic concept is that local ischemia augments delivery of NO species. While the present data are suggestive of this idea, the mechanism has to be shown more directly. What for example is the effect of acidic pH in vitro? What is the pO₂ in the NO patch implanted on non-infarcted sham rats? One would expect hypoxia to occur also under these conditions.

Reply: We thank the reviewer for their comments. According to previous studies, ischemic microenvironments were thought to be acidic because of necrosis [Sci Rep 2019;9:18427.] and extracellular release of H⁺[Proc Natl Acad Sci USA 2013;110:82-6.], which helped form the basis of our study about acidic-catalyzing NO generation from NO-patches [Nat Med 1995;1:804-9]. To provide direct evidence that low pH enhanced NO generation from NO-patches, we incubated the patches under different pH conditions and NO formation was measured by fluorescence intensity, using NO-specific probes. We observed that lower pH incubation strikingly promoted NO generation (**Suppl. Fig.9**). Moreover, hypoxia occurred due to the obstruction of blood flow. Following LAD ligation, cardiac partial pressure of oxygen (pO₂) decreased, indicating obvious hypoxia. While implantation of NO-patches elevated pO₂ levels in MI hearts, suggesting improved oxygen distribution and perfusion status. Implantation of NO-patches to non-infarcted hearts resulted in negligible changes to pO₂ in the myocardium (**Suppl. Fig. 10**).

2. The degree of protection is striking and a bit difficult to believe. How do the authors ensure equal injury sizes? Given the small number of animals, even minor differences in the proximity of LAD ligation can largely confound the results.

Reply: To ensure comparable severity of injury and injury size between groups, rats without pre-determined assignment received LAD ligation at a position 2 mm below the cross-junction of the left atrial appendage and the pulmonary conus, and patch implantation was performed randomly by a surgeon blind to the groups. Moreover,

baseline cardiac function measurements (Fig. 5a baseline) further confirmed that animals in each group had similar starting condition (Page 17, Lines 476-478).

3. *I did not find measures against bias such as blinding and randomisation procedures.*

Reply: We thank the reviewer for raising the important points above mentioned. In this study, the MI surgery, patch implantation, echo measurement as well as histological analysis were performed by researchers that were blind to the groups. We have added a section to provide further details on the blinding and randomization in the revised manuscript (Page 17, Lines 478-479, Page 19, Line 526, Page 21, Line 564, Page 23, Line 619).

4. *It would strengthen the study if key experiments e.g. in rats were reproduced by another group.*

Reply: I respectfully agree with you that repeating the experiments by other groups would reinforce our conclusions. If the reviewer and editor allow it, we would hope to perform such experimental repeats across multiple groups in future investigations, as opposed to the current study.

5. *Fig. 5D: Measurement of CM size by H&E staining is not state-of-the-art, please supplement with wheat germ agglutinin or dystrophin staining which better demarks cell borders.*

Reply: We thank the reviewer for this suggestion and have performed WGA staining and optimized the presentation in the revised manuscript (**Suppl. Fig. 14a** & Page 7, Lines 193-196).

6. *It is difficult to understand how NO should stimulate CM proliferation, which is extremely low in the postnatal heart and not or only very mildly higher in injured hearts (see consensus statement in Circulation 2017; PMID:28684531). I am not aware that NO has a convincing effect on CM proliferation (in contrast to interventions into the Hippo pathway). Thus, the marked effect shown in Fig. 5G and 7G is difficult to reconcile. Moreover, the measurement of Ki67 in histological sections is clearly not sufficient to proof CM proliferation. Please provide higher resolution confocal images of PH3 and aurora B kinase stainings and/or FACS staining for a CM marker such as actinin, troponin and Ki67 or PH3. It must also be differentiated between polyploidy and multinucleation and true cytokinesis, probably best by aurora B kinase staining.*

Reply: We thank the reviewer for their careful consideration and detailed comment regarding our CM proliferation data. As pointed out by the reviewer, it's well

established that interfering with the Hippo/YAP pathway can regulate CM proliferation, cardiac regeneration and repair [Nature 2017;550:260-4; Nature 2016;534:119-23; Development. 2013;140:4683-90.]. Despite the lack of a direct interaction of NO with the Hippo/Yap pathway, the roles of NO on cellular proliferation and cell cycling has been reported by different groups [J Am Coll Cardiol 2013;62:89-95; Am J Physiol Cell Physiol 2006;291:C1240-6.]. Through stimulation of the EGFR and PI3K/Akt pathway, NO was found to have functions in eliciting cell proliferation.

To strengthen our conclusions regarding NO regulation of CM proliferation, we performed the suggested confocal staining of Aurora B kinase (Z-stack), pH3 and Ki67 with the CM marker α -SA, and the results indicated that NO delivery by nitrate-functionalized patches enhanced CM proliferation after myocardial infarction (**Fig. 5 d-f** & Page 7-8, Lines 199-203).

7. For a therapy to be clinically applicable, effects of the NO patch would need to be shown in a model of chronic injury, e.g. implantation 4-8 weeks after MI. It is hard to imagine that patches were applicably in the settings of an acute MI.

Reply: In order to make our research more clinically applicable, we further introduced the NO-patches to chronic settings; 4 weeks after MI. The results indicated that treatment with NO-patches could sustain cardiac function and suppressed cardiac dilation, which further supported the protective/benefit roles of NO-patches for myocardial infarction. The results have been provided in the revised manuscript (**Fig. 6** & Page 8, Lines 210-215).

8. I did not find a COI statement.

Reply: The COI statement has been supplemented in the “Editorial policy checklist” uploaded as the supplementary file.

Reviewer #2 (Remarks to the Author):

In the work “Nitrate-functionalized patch confers cardioprotection and improves heart repair after myocardial infarction via local nitric oxide delivery” by Zhu et al., the authors have prepared a biodegradable polymer functionalized by nitrate as the NO donor for the treatment of cardiovascular disease. The authors have confirmed the NO release *in vivo* using fluorescence probes, and checked the effect on cardiac disease in pig. However, many similar works on the biodegradable polymer NO donors have been reported, such as “ACS Macro Lett. 2019, 8, 1552-1558”, “Polymer Bulletin 2018, 75, 2971–2985”, and “Biomater. Sci., 2013, 1, 625-632”. The authors have not demonstrated any advantages over these works. Consequently, this work should be rejected due to the lack of novelty.

Reply: We thank the reviewer for their comments. In fact, these three papers mainly focused on the development of the NO-releasing polymers without evaluating their potential in myocardial infarction treatment. Furthermore, following comprehensive comparison of the therapeutic efficacy of our NO-patches with others that have reported on the development of cardiac patches (principally in murine models; **Supplementary Material**), we can firmly conclude that the therapeutic efficacy of our NO-patches improved upon and was higher than the level of therapeutic effects observed in the previously published literature. Our results showed that these outcomes were demonstrably superior, when compared to investigations into other bioactive molecules for MI treatment (cytokines, genes, etc.), and this was attributable to the crucial therapeutic function of NO.

Additional comments:

1. *The detail characterizations for the polymer (PCL-ONO₂) and NO probe were necessary, therefore the ¹³C NMR and MS data should be provided.*

Reply: We thank the reviewer for this suggestion. ¹³C NMR and MS for PCL-ONO₂ and NO probe DAC-S have been performed. The results are included in the revised manuscript (**Suppl. Figs 2-3 and 5-7**).

2. *The NO-sensitive probe was used to acquire the NIR fluorescence images of infarcted myocardium by the response of the probe to NO. The response of the probe to NO which was released from the *in vitro* degradation of polymer (PCL-ONO₂), should be evaluated. In addition, why did the author choose three sites instead of more sites at myocardium to perform *ex vivo* NIR imaging (Fig. 2c)?*

Reply: According to your suggestion, NO generated from the *in vitro* degradation of PCL-ONO₂ was successfully tracked by using DAC-S probe, due to the high specificity and sensitivity.

In *ex vivo* imaging assays, four different planes of the heart gave an overall illustration of NO distribution across the whole heart.

3. How to assess the toxicity of PCL-ONO₂ in the treatment?

Reply: As has been well described, PCL is a biodegradable and biocompatible polymer that has been approved by the FDA in various biomedical applications, including drug delivery devices, sutures or adhesion barriers, and so on. The *toxicity* was evaluated by cell proliferation assay using CCK-8 kits, and the results did not show detectable differences between PCL-ONO₂ and pure PCL.

4. In page 4 and 5, the authors said “The quantitative data indicated that in the distal region, there was no detectable difference in NO levels among the various groups (Fig. 2c), but in the infarcted myocardium, NO levels were evidently enhanced in the MI group due to endogenous NO production by activated macrophages.” This meaning that fig 2d, the high concentration of NO in MI+NO-patch group was produced from the activated macrophages not from the PCL-ONO₂ system. This is contradictory, please give the reasonable explanations.

Reply: We are sorry for the ambiguous statement. In fact, our results demonstrated that ischemia induction led to the enhancement of detectable NO signal in both MI and MI+PCL-patch groups, compared to the sham group. This finding was because of the polarization of macrophages which was further confirmed by *in vitro* cell experiments (**Suppl. Fig. 4**). There was a more pronounced enhancement observed in the MI+NO-patch group, and this was significantly higher than MI and

MI+PCL-patch groups ($p < 0.0001$ or 0.001). The source of the additional NO detected was from the PCL-ONO₂. The relevant statement has been rewritten in the revised manuscript (Page 4-5, Lines 116-119).

Reviewer #3 (Remarks to the Author):

Comments

1) General comment: The entire manuscript needs English proof correction (see examples in the abstract below).

Reply: The language has been polished by the **Springer Nature Author Services** (verification code F4E8-97D7-F1A0-91CD-04EP). The certificate has been uploaded as the supplementary file.

2) General comment: I understand the rationale to use these nitrate patches that can be implanted nearby the heart, where infarct damage is most severe. This especially allows to reach higher local doses of the nitrate, which otherwise cannot be given at very high oral doses as this would cause serious hypotension with life-threatening loss of blood pressure. I am not sure, whether this most important benefit of a nitrate-containing biomaterial that can be implanted, was sufficiently addressed in the present manuscript. In the Introduction a respective rationale for using the drug-containing patch, rather than using oral administration of the nitrate was chosen in the present work. In the Discussion, please add the pros and cons of the nitrate-biomaterial versus oral administration.

Reply: We thank the reviewer for their comment. Necessary discussion on this point has been included in the revised manuscript (Pages 12-13, Lines 343-348), and reads as follows:

Compared to organic nitrates, nitrate-functionalized patches generate NO via an extracellular stepwise biotransformation. As a result, local transformation of NO remarkably abolishes adverse effects, including serious hypotension caused by the vasodilation of systemic vasculature when higher doses of organic nitrates are administered via oral or transdermal pathways. Local NO delivery can further provide non-hemodynamic effects on the ischaemic myocardium, including cardioprotection and immunomodulation.

3) General comment: NO-release from biopolymers was already used in the drug-eluting stents (DES), which play an important role for prevention of MI but also first-line treatment after MI. This important point is not mentioned at all in the present work, which represents a major weakness and also implies that the authors did not read the essential background on NO and nitrates usage against MI damage in the clinics. I think that NO-eluting stents did not prove to be superior to bare metal stents – what would this mean for your proposed concept here?

Reply: We are aware that NO-releasing coatings have been employed in the development of drug-eluting stents [Proc Natl Acad Sci USA 2020;117:16127-37; Research (Wash D C) 2020:9203906.] for coronary heart diseases, and that some of

these have entered into pre-clinical trials. However, based on the limited data that has been reported up until now, the therapeutic efficacy of NO-releasing stents is far from satisfactory because of intimal hyperplasia representing the major adverse event following the complete release of loaded NO-drug in longer-term studies [Yonsei Med J 2002; 43:242-51]. In light of this report, it was long-term NO delivery that proved to be a main challenge. This is because existing drug-eluting systems have a finite depot of drug and will lose therapeutic function once the depot is depleted [Proc Natl Acad Sci USA 2014; 111:12722-7.]. Instead, surface modification of bare stents by immobilizing relevant enzymes to catalyze the bioactivation of endogenous NO donors (S-nitrosothiol or nitrate) may represent a new paradigm for the future of NO-releasing stents.

4) General comment: Also nitrate tolerance, which represents a major limitation of the clinical use of organic nitrates, was not mentioned at all, which represents a major weakness and also implies that the authors did not read the essential background on NO and nitrates usage against MI damage in the clinics. The authors should introduce and discuss this major adverse side effect of nitrate therapy (for review see PMID: 21576678, PMID: 23864131, PMID: 26261901) as it may also represent another strength of the here proposed NO-releasing patch that can be implanted as I assume that implantation of the nitrate-containing patch is not subject to nitrate tolerance, which may represent a major advantage besides the higher local concentrations of NO as compared with oral nitrate administration.

Reply: Thanks for your valuable suggestion. Necessary discussion on this point has been included in the revised manuscript, and reads as follows (Page 12, Lines 321-337):

Tolerance represents a major limitation of some organic nitrates utilized in the clinic, especially glyceryl trinitrate (GTN). GTN is the most commonly utilised treatment for various ischemic and congestive cardiac diseases⁵³⁻⁵⁵, although there are also organic nitrates (such as pentaerythritol tetranitrate; PETN) that are devoid of the effects of tolerance emergence^{56, 57}. Nitrate tolerance is a complex phenomenon and yet to be clearly elucidated. In general, the typical consensus is that nitrate tolerance is mainly caused by oxidative inactivation of the reductase activity of aldehyde dehydrogenase-2 (ALDH2), which results in decreased GTN bioactivation, and the resultant diminished levels of nitrite (NO₂⁻) and nitric oxide (NO). Recently, Mochly-Rosen et al. have also reported that activators of ALDH2 such as Alda-1 may help to inhibit cardiac injury induced by GTN tolerance^{58, 59}.

In contrast, as a macromolecular nitrate, PCL-ONO₂ demonstrates a specific biotransformation pathway that is different from that of small molecule drugs. First, polymers undergo hydrolysis to release nitrate anions. Then, the generated nitrate anions are reduced to release NO via the nitrate-nitrite-NO sequential pathway. Since the phenomenon of tolerance is not exhibited with the consumption of inorganic nitrates/nitrites⁶⁰, it is reasonable to speculate that PCL-ONO₂ developed in this study is free from tolerance, although relevant experiments are required to validate this

hypothesis.

The important references suggested by the reviewer (PMID: 21576678, PMID: 23864131, PMID: 26261901) have been cited in the revised manuscript.

5) *General comment: The authors briefly mention organic nitrate biotransformation/bioactivation, which is a crucial step in the pharmacological action (NO release) of a given organic nitrate. The authors should make themselves familiar with the routes of organic nitrate bioactivation (for review see PMID: 21576678, PMID: 23864131, PMID: 26261901). As they correctly mention, the nitrate-containing biomaterial cannot cross the cell membrane (as low molecular weight nitrates do) and reach the place of bioactivation (usually an enzyme such as aldehyde dehydrogenase, P450s or potentially xanthine oxidoreductase). Instead, the authors propose spontaneous hydrolysis and inorganic nitrate release from the nitrate-containing biomaterial. Then however, this method would compete with oral administration of inorganic nitrate (which can be administrated at high doses, e.g. via vegetables such as beetroot or spinach). What is then the advantage over oral inorganic nitrate administration, which also proved beneficial to prevent MI or ameliorate MI damage? This needs to be explained in detail and also the bioactivation of organic and inorganic nitrates should be better explained.*

Reply: The transition of nitrate-functionalized PCL patch proceeds through multiple steps that is different from the biotransformation of organic nitrates. Since the patch was transplanted in the infarcted heart, higher local concentration of nitrate/nitrite could be realised. We respectively agree with you that oral administration of inorganic nitrate may be an alternative strategy to prevent or ameliorate MI that has been reported before. The main disadvantage lies in the systemic release of NO, with higher doses causing serious hypotension. At the same time, the majority of orally administered inorganic nitrates results in a larger proportion of NO generated in blood vessels and providing vasodilatory effects instead of the desirable cardioprotection. Necessary discussion on this point has been included in the revised manuscript (Pages 12-13, Lines 343-348), and the bioactivation of organic and inorganic nitrates has been explained in detail (Pages 9, Lines 248-252, Pages 10, Lines 261-264).

6) *General comment: As inorganic nitrate should be the “active” species released from the nitrate-containing biomaterial, the authors should introduce and discuss the current knowledge on inorganic nitrate bioactivation and reported beneficial effects in MI and other heart disease models (for review see PMID: 26265312, PMID: 18167491).*

Reply: Thanks for your valuable suggestion. We have provided relevant discussion on the inorganic nitrate bioactivation in the revised manuscript, as follows (Page 9-10, Lines 253-260):

Currently, there is accumulating evidence demonstrating that inorganic nitrates represent an important reservoir for NO in the body [4,45]. Under specific conditions, inorganic nitrates can be metabolized to generate NO through the nitrate-nitrite-NO reduction pathway. Nitrates are found in considerable amounts in the diet, and many green vegetables such as spinach or beetroot are particularly rich in nitrates. Recently, supplementation of inorganic nitrates has been attempted by different groups with the aim of providing therapy for cardiovascular diseases. Promising outcomes have been reported, including improved revascularization in chronic ischemia³⁵, and stabilized atherosclerotic plaques³⁶.

These two important references (PMID: 26265312, PMID: 18167491) have been cited in the revised manuscript.

7) *General comment: Overload with NO is also not good and termed “nitrosative stress”. Did the authors observe any signs of nitrosative stress in their model (e.g. dramatic increase in S-nitrosated proteins, some SNO modification cause loss of function in enzymes)?*

Reply: In general, overdose of NO would cause “nitrosative stress”, because NO reacts rapidly with other radicals, such as superoxides, resulting in the formation of other reactive nitrogen species (RNS), such as peroxynitrite (ONOO⁻). Peroxynitrite can induce protein nitration, cytotoxicity, and plays important roles in the development of cardiovascular pathophysiology. In this study, the peroxynitrite level in the heart tissue was evaluated and it did not show detectable variation after the NO patch treatment, thereby suggesting that nitrosative stress did not result from the treatments.

8) *General comment: Whereas the presented data are impressive, the authors need to present a head-to-head comparison of protective effects of the nitrate-containing patch versus oral or intravenous organic nitrate administration versus inorganic nitrate treatment during ischemia/reperfusion. How is infarct size and cardiac function affected by these three treatment protocols in the same infarct model? There are also organic nitrates that are devoid of nitrate tolerance and therefore could be*

used at high dose (e.g. PETN, see PMID: 11527645, PMID: 24071762).

Reply: We appreciate the suggestion to do a head-to-head comparison study - as this is very important to demonstrate the advantages and disadvantages of nitrate-functionalized patches, compared with both organic and inorganic nitrates that have been reported previously.

We have included a discussion addressing this point in the revised manuscript (Page 13, Lines 350-353).

These two important references (PMID: 11527645, PMID: 24071762) have been cited in the revised manuscript.

9) General comment: Previously, it was shown that already improvement of organic nitrate bioactivation and prevention of nitrate tolerance may be sufficient to conserve the anti-ischemic effects and protection by organic nitrates such as nitroglycerin in a MI model (PMID: 18787169, PMID: 22049071). How would your nitrate-containing patch compete with this pharmacological activation of nitroglycerin bioactivating enzyme ALDH-2 by Alda-1?

Reply: Thanks for your interesting question. Recently, Mochly-Rosen et al. reported that activators of ALDH2 such as Alda-1 may help to inhibit cardiac injury induced by GTN tolerance. In contrast, nitrate-functionalized PCL does not require additional activators, because the process of biotransformation is different from that of classic organic nitrates.

Necessary discussion on this point has been included in the revised manuscript (Page 12, Lines 321-337), and these two important references (PMID: 18787169, PMID: 22049071) have been cited in the revised manuscript.

10) General comment: I have seen much more impressive improvements by some treatment protocols in MI animal models in the past than those survival data presented in figure 6. Therefore, it is really important to compare head-to-head different treatment protocols (see my suggestions above, e.g. oral organic nitrate, inorganic nitrate, Alda-1) to demonstrate the advantage of the here presented method over those previously reported.

Reply: We appreciate the suggestion to do a head-to-head comparison study - as this is very important to demonstrate the advantages and disadvantages of nitrate-functionalized patches, compared with both organic and inorganic nitrates that have been reported previously.

We have included a discussion addressing this point in the revised manuscript (Page 13, Lines 350-353).

11) Abstract: „wherein the nitrate pharmacophores were covalently bonded to biodegradable polymers“ – „pharmacophores“ is unclear meaning, better use

„pharmacologically-active groups“ or „pharmacological functional groups“. Also „bonded“ sounds odd, better use „bound“.

Reply: Thanks for your suggestion, and they have been corrected in the revised manuscript.

12) Abstract: „infarcted zone“ should be „infarcted area“ or „infarcted tissue“.

Reply: They have been corrected accordingly in the revised manuscript.

13) Abstract: “with therapeutic mechanism different” – sounds odd, better use “by therapeutic strategies different”.

Reply: It has been corrected accordingly in the revised manuscript.

14) Other specific concerns: The number of animals in many of the measured parameters is very low (n=3), which hardly allows a full statistical analysis.

Reply: Some *in vivo* experiments have been repeated and the number of animals for each experiment is now greater than or equal to 5, for all data presented in the revised manuscript.

15) Other specific concerns: why was survival only tested in the pig model with very low amount of animals? I think with the rodents it would have been easier to have large animal numbers for the survival studies.

Reply: Thanks for your comment. Since we recorded high survival rate of rats after receiving MI procedures, and there is no evident difference detected between two groups, therefore we didn't show survival tests in rats. Whilst in pigs, the survival rate was significantly improved by NO-patch implantation.

16) Animals: Please provide the ethical permission number for the different animal experiments.

Reply: They have been provided in the revised manuscript (Page 18, Line 482 and Page 22, Line 600).

17) Animals: What kind of analgesia was administrated to the animals after MI surgery. As they feel pain afterwards, the international animal protection guidelines recommend to use analgetic drugs after MI (LAD ligation) surgery.

Reply: In this study, Buprenex (0.01 mg/kg) and Carprofen (5.00 mg/kg) were administrated to relief pain and distress after procedures (Page 18, Lines 495-496).

18) Animals: what method was used for sacrifice of the animals and what anesthesia was used for that?

Reply: Rats were sedated and anesthetized by isoflurane inhalation, followed by deep anesthesia via I.P. injection of chloral hydrate (350 mg/kg) (confirmed by toe pinch) (Page 21, Lines 564-566). The rats were sacrificed three days after surgery through CO₂ inhalation. (Page 21, Line 588). Pigs were anesthetized by inhalation of isoflurane, and pentobarbital solution (390mg/mL, 1mL/5kg) was injected to euthanasia. (Page 24, Lines 645-646)

Reviewer #4 (Remarks to the Author):

In this manuscript, the authors developed a nitrate conjugated PCL patch for treating ischemic heart diseases. Nitrate conjugated PCL patch was hydrolyzed under inflammatory micro-environment after myocardial infarction, nitrate ion was released and further converted to nitric oxide. In vivo data demonstrated cardioprotection effects by using NO-patch on rat and porcine myocardial infarction model. However, the authors did not provide compelling rationale for utilizing cardiac patch on therapeutic molecules delivery.

Reply: We thank the reviewer for their comments. Following comprehensive comparison of the therapeutic efficacy of our NO-patches with others that have reported on the development of cardiac patches (principally in murine models; **Supplementary Material**), we can firmly conclude that the therapeutic efficacy of our NO-patches improved upon and was higher than the level of therapeutic effects observed in the previously published literature. Our results showed that these outcomes were demonstrably superior, when compared to investigations into other bioactive molecules for MI treatment (cytokines, genes, etc.), and this was attributable to the crucial therapeutic function of NO.

Specific comments:

1. Using cardiac patch can provide scaffolding for culturing therapeutic cells (e.g., cardiac cells) and be used as an implantation to repair injured heart. As for covalently conjugated drug delivery platform, cardiac patch requires more time to release the therapeutic molecules. Compared to nano/microparticles system, what is the main advantage to use cardiac patch to deliver therapeutic molecules?

Reply: Compared to nano/microparticles system, cardiac patch not only acts as a carrier for the delivery of therapeutic molecules, it can also provide biomechanical support for damaged myocardium. Acellular cardiac patches are another therapy that treat MI by restricting ventricular dilatation, preventing adverse left ventricular remodelling and improving contractile function that is directly attributable to mechanical support by elastic polymers^{22,23}. (Page 3, Lines 68-70)

As a result, more desirable therapeutic effects could be observed through optimizing cardiac patch in terms of both mechanical performance as well as therapeutic molecules.

2. Although the water contact angle slightly decreased after nitrate conjugation, NO-patch was still relatively hydrophobic (water contact angle was ~120°). It is unclear how this property would be favorable for the tissue microenvironment.

Reply: Thanks for your observation. Although the hydrophilicity has been enhanced in the nitrate-functionalized PCL compared to pure PCL, the patch is still hydrophobic. As an epicardial patch, it mainly acts as drug delivery carrier to release NO locally,

and we speculate that the patch structure itself did not participate directly in the regeneration of myocardium. As a result, the surface property can satisfy the therapeutic requirement, i.e. drug carrier. The relevant statement has been rewritten accordingly in the revised manuscript (Page 4, Lines 97-98).

3. The subcutaneous NO release could only last for 120 minutes (figure 1f). It is quite short-lived. How does this translate into therapeutic benefits?

Reply: In this study, the patches were implanted subcutaneously in rats, and following their explant, the residual NO_x within the patch was determined by chemiluminescence. The duration for the measurement was 120 minutes (Fig. 1 f). Our results show that the patch released NO in a controlled manner, and the residual NO_x exceeded 26.6% after 28 days of implantation (Fig. 1 g).

4. The authors described that the elevated NO levels in the infarcted area was due to endogenous NO production via activated macrophages. However, there was no evidence showing the increased NO was produced by macrophages.

Reply: The generation of NO by activated macrophages has been observed by using NO-specific fluorescence probe. The results have been provided in the revised manuscript (**Suppl. Fig. 4**).

5. There was a gap between NO-patch and myocardium in supplementary figure 9. Would this gap prevent leukocytes infiltration into NO-patch?

Reply: Thanks for your observation. Actually, the patch was closely adhered to the myocardium after the implantation. The gap was generated during the process of histological analysis, that is, paraffin sectioning; the patch was easily separated from the heart tissue during the preparation for sectioning.

6. It is not clear what is 5G in figure 2b.

Reply: It's the unit (scale bar) for the EPR spectrum.

7. It is difficult to compare cardiomyocyte size in figure 3d. More representative images are needed.

Reply: We have optimized the results with WGA staining that can clearly show the border of cardiomyocytes, and the cross section areas were measured to quantify the differences (**Suppl. Fig.14a**).

Reviewers' comments:

Reviewer #1 (Remarks to the Author):

The authors have addressed some of my concerns, but inconsistencies in the new data in Fig. 5 and 6 rather aggravate my concerns.

Fig. 5: It is well known that Ki67 staining persists during the entire mitosis while PH3 staining only in a short window. It is therefore disturbing that PH3-positivity was determined at such high value (up to 4 CM/HPF) while Ki67+ was ~2 CM/HPF. Similarly, aurora B kinase staining is expected at exceedingly low levels even in case of relevant CM proliferation, but here the authors find up to 4 CM/HPF. This does not make sense and casts doubts on the validity of measurements. Moreover, this reviewer cannot identify any typical aurora B kinase midbody staining, the accepted sign of cell division (PMID: 30355161) in the example shown in Fig. 5.

Fig. 6: The examples shown in this figure suggest a major effect of the NO patch on infarct size. Quantification indicates a NO patch-associated reduction in infarct size from ~40% to <30%, a major effect, but much smaller than that shown in the example. Overall, a reduction in infarct size by NO is a surprising effect and does not align with the lack of functional improvement shown in Fig. 6D,E.

The authors argue in their rebuttal that NO has been described to affect cell proliferation and refer mainly to a review article from 2013, where Ignarro and colleagues describe anti-proliferative and pro-proliferative effects. However, nowhere in this review (or original literature this reviewer is aware of), a robust pro-proliferative effect of NO (donors) on adult, post-mitotic cardiomyocytes (CM) has been described. This has been an extremely intensely studied question over more than a decade and the consensus is that proliferation is extremely rare after birth. The few interventions that describe convincing CM proliferation stimulation with agrin, ErbB2, various miRNAs or direct Yap overexpression all relate in some way to the Hippo pathway, but to the best of the knowledge of this reviewer, not to NO or cGMP.

Reviewer #3 (Remarks to the Author):

Comments

1) ok.

2) ok.

3) General comment: NO-release from biopolymers was already used in the drug-eluting stents (DES), which play an important role for prevention of MI but also first-line treatment after MI. This important point is not mentioned at all in the present work, which represents a major weakness and also implies that the authors did not read the essential background on NO and nitrates usage against MI damage in the clinics. I think that NO-eluting stents did not prove to be superior to bare metal stents – what would this mean for your proposed concept here?

Follow-up comment: The response is appreciated but should be incorporated into the MS discussion, as this is exactly the critical issue to deliver NO at the right place at the right time. Also DES have these problems. Please also consider a work on AAV-dependent delivery of iNOS to the stent leading to sustained generation of NO (PMID: 28832561).

4) ok.

5) General comment: The authors briefly mention organic nitrate biotransformation/bioactivation, which is a crucial step in the pharmacological action (NO release) of a given organic nitrate. The authors should make themselves familiar with the routes of organic nitrate bioactivation (for review see PMID: 21576678, PMID: 23864131, PMID: 26261901). As they correctly mention, the nitrate-containing biomaterial cannot cross the cell membrane (as low molecular weight nitrates do) and reach the place of bioactivation (usually an enzyme such as aldehyde dehydrogenase, P450s or potentially xanthine oxidoreductase). Instead, the authors propose spontaneous hydrolysis and inorganic nitrate release from the nitrate-containing biomaterial. Then however, this method would compete with oral administration of inorganic nitrate (which can be administered at high doses, e.g. via vegetables such as beetroot or spinach). What is then the advantage over oral inorganic nitrate administration, which also proved beneficial to prevent MI or ameliorate MI damage? This needs to be explained in detail and also the bioactivation of organic and inorganic nitrates should be better explained.

Follow-up comment: I think it is not possible to overdose inorganic nitrate by oral uptake to cause hypotension as before, severe methemoglobinemia would occur. Therefore, the explanation of the authors is odd. Please find better explanation for potential advantages of your NO-releasing patch as compared to inorganic nitrate or nitrite. Also do not use “Nitrates” when you talk about inorganic nitrate as there are no different nitrate forms, only one.

6) ok.

7) General comment: Overload with NO is also not good and termed “nitrosative stress”. Did the authors observe any signs of nitrosative stress in their model (e.g. dramatic increase in S-nitrosated proteins, some SNO modification cause loss of function in enzymes)?

Follow-up comment: “Nitrosative stress” is different from peroxynitrite-mediated “nitro-oxidative” stress that leads to protein tyrosine nitration. The authors mixed-up these concepts. “Nitrosative stress” causes S-nitrosothiol formation, e.g. GSNO and S-nitrosylated proteins such as albumin, caspases, complex I, cyclophilin D and many more, that can be measured in plasma or tissues. So it would be appreciated if the authors could show markers of “nitrosative stress”.

Regarding the figure on peroxynitrite formation that they show - I can neither find the figure nor the method description in the MS. How was it measured, please explain the method in very detail (just mentioning that it was a commercial kit will not suffice). The reason is that I am not aware of any specific method for detection of peroxynitrite and would be interested to know more about the presented assay.

8) General comment: Whereas the presented data are impressive, the authors need to present a head-to-head comparison of protective effects of the nitrate-containing patch versus oral or intravenous organic nitrate administration versus inorganic nitrate treatment during ischemia/reperfusion. How is infarct size and cardiac function affected by these three treatment protocols in the same infarct model? There are also organic nitrates that are devoid of nitrate tolerance and therefore could be used at high dose (e.g. PETN, see PMID: 11527645, PMID: 24071762).

Follow-up comment: Without the head-to-head comparison, the presented data on the protective effect of your NO-releasing patch against MI damage are almost useless because without the direct comparison to other treatments in the same MI damage model one cannot judge whether the patch method is superior or not. Please add the head-to-head comparison at least investigating some key parameters in the rat model.

9) General comment: Previously, it was shown that already improvement of organic nitrate bioactivation and prevention of nitrate tolerance may be sufficient to conserve the anti-ischemic effects and protection by organic nitrates such as nitroglycerin in a MI model (PMID: 18787169, PMID: 22049071). How would your nitrate-containing patch compete with this pharmacological activation of nitroglycerin bioactivating enzyme ALDH-2 by Alda-1?

Follow-up comment: Answer almost fine. Please add a critical discussion on the advantage of your NO-releasing patch over the nitroglycerin/Alda-1 combination.

10) General comment: I have seen much more impressive improvements by some treatment protocols in MI animal models in the past than those survival data presented in figure 6. Therefore, it is really important to compare head-to-head different treatment protocols (see my suggestions above, e.g. oral organic nitrate, inorganic nitrate, Alda-1) to demonstrate the advantage of the here presented method over those previously reported.

Follow-up comment: see my new comment #8.

11) ok.

12) ok.

13) ok.

14) ok.

15) Other specific concerns: why was survival only tested in the pig model with very low amount of animals? I think with the rodents it would have been easier to have large animal numbers for the survival studies.

Follow-up comment: Maybe in the future mice would be the better choice as they show appreciable mortality after LAD ligation induced MI. I still think that doing a survival study with less than 10 animals cannot yield very robust data.

16) ok.

17) ok.

18) ok.

Reviewer #4 (Remarks to the Author):

The revision addressed most of my concerns. However, the revision added a justification of the patch " Acellular cardiac patches are another therapy that

treat MI by restricting ventricular dilatation, preventing adverse left ventricular remodelling and improving contractile function that is directly attributable to

mechanical support by elastic polymers (Page 3, Lines 68-70) " This implies the author was claiming PCL was elastic. I don't think they claimed so, but this could lead to confusion. PCL is not an elastomer and I suggest the authors to add a line to clarify that.

Reviewer #5 (Remarks to the Author):

The authors addressed most of the questions of the reviewers. The main concern of reviewer 2 is the novelty of the work and I strongly agree with him or her. The biodegradable polymeric NO donor used in this work was structurally similar to the reported ones and the mechanism of NO releasing was completely same. Although the authors emphasized the improved therapeutic efficacy, no theoretical reason was provided to explain why the polymer was better than the reported ones. On the basis of lack of theoretical support, it is difficult to judge the therapeutic efficacy with such a small sample amount.

Reviewers' comments:

Reviewer #1 (Remarks to the Author):

The authors have addressed some of my concerns, but inconsistencies in the new data in Fig. 5 and 6 rather aggravate my concerns.

Fig. 5: It is well known that Ki67 staining persists during the entire mitosis while PH3 staining only in a short window. It is therefore disturbing that PH3-positivity was determined at such high value (up to 4 CM/HPF) while Ki67+ was ~2 CM/HPF. Similarly, aurora B kinase staining is expected at exceedingly low levels even in case of relevant CM proliferation, but here the authors find up to 4 CM/HPF. This does not make sense and casts doubts on the validity of measurements. Moreover, this reviewer cannot identify any typical aurora B kinase midbody staining, the accepted sign of cell division (PMID: 30355161) in the example shown in Fig. 5.

Reply: We thank the reviewer for pointing this out. We did interpret the datasets according to the measurements, and the reactivities of the antibodies were confirmed by employing negative control staining. In our trials, Ki67⁺ CMs were rarely detected, whereas PH3⁺ proliferating nucleus were more commonly observed. Given the validity of antibodies, even though the result goes against the theory that Ki67 persists during the entire mitosis and that PH3 presents only in a limited period, the relative tendency as shown in Figures 5d and 5f was rational. Aurora B kinase was expressed during the mitosis with different distribution patterns in different phases, as in prophase, dot-like distribution across the whole nucleus (similar pattern as shown in our results) were presented. Therefore, the Aurora B staining shown in Fig 5e was valid. However, since no midbody that indicated authentic mitosis was observed and there was a conflict regarding actual cell counts of Ki67 vs PH3, we'd like to remove the data from the revised manuscript. We have added relevant discussion regarding to this point as follows (Page 14, Lines 403-407):

In this study, we detected increased numbers of Ki67⁺, PH3⁺, Aurora B⁺ cardiomyocytes after NO-patch treatment, however, there is no actual division as shown by the formation of midbody observed in Aurora B staining, therefore further studies on the relationship of NO with cardiomyocyte mitosis and cardiac regeneration are needed to reach a consensus conclusion.

Figure 1. Validation of antibody reactivity. To confirm the reactivities of the antibody (Ki67, pH3 and Aurora B kinase), negative control(5% normal goat serum) and antibodies at listed dilution(in 5% normal goat serum) were used to incubate section overnight at 4°C. The species of all three antibodies is rabbit. And secondary

Alexa Fluoro-488 conjugated goat anti-rabbit antibody was used for detection.

Fig. 6: The examples shown in this figure suggest a major effect of the NO patch on infarct size. Quantification indicates a NO patch-associated reduction in infarct size from ~40% to <30%, a major effect, but much smaller than that shown in the example. Overall, a reduction in infarct size by NO is a surprising effect and does not align with the lack of functional improvement shown in Fig. 6D,E.

Reply: We thank the reviewer for their observation. In fact, the infarct size shown in Fig. 6c is the mean of 6 different samples. Unfortunately, upon revision, the sample (NO-patch 3) selected for Fig. 6b was not representative of the actual level. It has been replaced by another sample (NO-patch 5) with infarct size more representative to the mean values calculated in the revised manuscript.

Figure 2. To calculate infarct size, the fibrotic scar was filled in blue, and accordingly, the infarct area and left ventricular (LV) area were measured. The values in the table were shown in arbitrary unit.

Sample code	Infarct area	LV area	% (Infarct/LV)
NO-patch-1	110407	371978	29.7
NO-patch-2	72072	222147	32.4
NO-patch-3	35974	144585	24.9
NO-patch-5	76795	288069	26.7
NO-patch-5	60902	198303	30.7
NO-patch-6	54008	185417	29.1

Indeed, the LV-EF has been significantly ($p < 0.05$) enhanced after NO-patch treatment compared with that before treatment. It can be observed from the raw data we have submitted before, shown below for your convenience (Source Data file of NCOMMS-20-11768A).

Fig5d										
Baseline(MI-1M)					After treatment					
Sham	72.5514	69.76755	70.5278	65.61675	64.918	64.62426	70.69837	70.67137	68.96885	68.68984
MI	31.28784	26.07484	30.53726	32.40342	31.73721	27.4773	29.10824	24.99729	31.52928	25.53447
PCL-patch	25.30475	31.74015	30.6154	28.50122	26.19581	26.46306	27.32565	28.51451	26.8312	24.93821
NO-patch	33.97331	24.70869	34.96364	31.2036	25.62388	26.72672	34.78539	33.15184	33.57302	32.62123
Fig5e										
Baseline(MI-1M)					After treatment					
Sham	43.289	40.50279	41.23699	37.28362	36.80104	36.73996	41.77363	41.5404	38.34197	40.07051
MI	15.55794	12.73123	15.14591	16.12542	15.81741	13.47826	14.42987	12.21996	15.76306	12.4871
PCL-patch	12.32279	15.81197	15.20591	13.99776	12.79826	12.94498	13.4375	14.09602	13.19018	12.1775
NO-patch	17.04423	12.00873	17.62208	15.48677	12.47216	13.09013	17.46385	16.61409	16.82744	16.2891
Fig5d Two-way ANOVA										
Baseline(MI-1M)	Mean Diff	95.00% CI of diff	Significant?	Summary	Adjusted P Value	Fig5e Two-way ANOVA				
Sham	-0.5613	-4.946 to 3.784	No	ns	0.6947	Sham	-0.5895	-3.287 to 2.108	No	ns
MI	1.93	-2.435 to 6.295	No	ns	0.6932	MI	0.9778	-1.72 to 3.676	No	ns
PCL-patch	1.413	-2.952 to 5.778	No	ns	0.8733	PCL-patch	0.7152	-1.983 to 3.413	No	ns
NO-patch	-4.391	-8.756 to -0.0265	Yes	*	0.0481	NO-patch	-2.407	-5.105 to 0.2911	No	ns

Further analyses also indicate that consistent with the reduction of cardiac infarct size, there was a detectable difference in cardiac function, including LV-EF and LV-FS, between NO-patch group and PCL-patch as well as MI groups 4 weeks post-implantation. We apologize for not indicating the *p* values and they have been added in the revised manuscript.

The authors argue in their rebuttal that NO has been described to affect cell proliferation and refer mainly to a review article from 2013, where Ignarro and colleagues describe anti-proliferative and pro-proliferative effects. However, nowhere in this review (or original literature this reviewer is aware of), a robust pro-proliferative effect of NO (donors) on adult, post-mitotic cardiomyocytes (CM) has been described. This has been an extremely intensely studied question over more than a decade and the consensus is that proliferation is extremely rare after birth. The few interventions that describe convincing CM proliferation stimulation with agrin, ErbB2, various miRNAs or direct Yap overexpression all relate in some way to the Hippo pathway, but to the best of the knowledge of this reviewer, not to NO or cGMP.

Reply: We respectfully agree with the reviewer that adult heart has extremely low rate of proliferation. Since the current results fail to support the proliferation of cardiomyocytes sufficiently in terms of lack of midbody (Aurora B staining) and no observed regeneration in the chronic model, the effect of NO on CM proliferation could not be fully supported and is not presented in the revised manuscript. Instead, we focus on the cardio-protective, pro-angiogenic and immune-modulatory effects provided by NO delivery in the present study.

A range of additional experiments has been performed in order to illustrate the unique biotransformation mechanism that is different from classical organic nitrate drugs as well as inorganic nitrate and provide theoretical support for the improved therapeutic efficacy in the present study (Figure 3, Supplementary Fig 9 and 11) (Page 5, Lines 123-148).

Reviewer #3 (Remarks to the Author):

Comments

1) ok.

2) ok.

3) *General comment: NO-release from biopolymers was already used in the drug-eluting stents (DES), which play an important role for prevention of MI but also first-line treatment after MI. This important point is not mentioned at all in the present work, which represents a major weakness and also implies that the authors did not read the essential background on NO and nitrates usage against MI damage in the clinics. I think that NO-eluting stents did not prove to be superior to bare metal stents – what would this mean for your proposed concept here?*

Follow-up comment: The response is appreciated but should be incorporated into the MS discussion, as this is exactly the critical issue to deliver NO at the right place at the right time. Also DES have these problems. Please also consider a work on AAV-dependent delivery of iNOS to the stent leading to sustained generation of NO (PMID: 28832561).

Reply: We thank the reviewer for this suggestion and have now included the relevant discussion in the revised manuscript as follows (Page 14, Lines 392-402). This important reference (PMID: 28832561) have been cited in the revised manuscript.

NO-releasing coatings have been employed in the development of drug-eluting stents⁹,¹⁰ for coronary heart diseases, and that some of these have entered into pre-clinical trials. However, based on the limited data that has been reported up until now, the therapeutic efficacy of NO-releasing stents is far from satisfactory because of intimal hyperplasia representing the major adverse event following the complete release of loaded NO-drug in longer-term studies⁶³. In light of this report, it was long-term NO delivery that proved to be a main challenge. This is because existing drug-eluting systems have a finite depot of drug and will lose therapeutic function once the depot is depleted⁶⁴. Instead, surface modification of bare stents by immobilizing relevant enzymes to strengthen endogenous NO generation⁶⁵ or catalyze the bioactivation of endogenous NO donors¹⁰, such as S-nitrosothiol, may represent a new paradigm for the future of NO-releasing stents.

4) ok.

5) *General comment: The authors briefly mention organic nitrate biotransformation/bioactivation, which is a crucial step in the pharmacological action (NO release) of a given organic nitrate. The authors should make themselves familiar with the routes of organic nitrate bioactivation (for review see PMID: 21576678, PMID: 23864131, PMID: 26261901). As they correctly mention, the nitrate-containing biomaterial cannot cross the cell membrane (as low molecular weight nitrates do) and reach the place of bioactivation (usually an enzyme such as*

aldehyde dehydrogenase, P450s or potentially xanthine oxidoreductase). Instead, the authors propose spontaneous hydrolysis and inorganic nitrate release from the nitrate-containing biomaterial. Then however, this method would compete with oral administration of inorganic nitrate (which can be administered at high doses, e.g. via vegetables such as beetroot or spinach). What is then the advantage over oral inorganic nitrate administration, which also proved beneficial to prevent MI or ameliorate MI damage? This needs to be explained in detail and also the bioactivation of organic and inorganic nitrates should be better explained.

Follow-up comment: I think it is not possible to overdose inorganic nitrate by oral uptake to cause hypotension as before, severe methemoglobinemia would occur. Therefore, the explanation of the authors is odd. Please find better explanation for potential advantages of your NO-releasing patch as compared to inorganic nitrate or nitrite. Also do not use "Nitrates" when you talk about inorganic nitrate as there are no different nitrate forms, only one.

Reply: We thank the reviewer for this suggestion and the relevant explanation has been revised as follows (Page 10-11, Lines 287-293).

On the other hand, *in vivo* transformation efficiency of inorganic nitrate is influenced by many factors, including the tissue, the pH, oxygen tension and redox status⁴³. In addition, a major health concern is that higher doses of nitrite derived from dietary nitrate may cause severe methemoglobinemia. It may also lead to an increased cancer risk due to the formation of the carcinogen substances N-nitrosamines from ingested nitrate and nitrite⁴⁴. As a result, the nitrate level in drinking water is strictly regulated in many countries⁴.

6) ok.

7) *General comment: Overload with NO is also not good and termed "nitrosative stress". Did the authors observe any signs of nitrosative stress in their model (e.g. dramatic increase in S-nitrosated proteins, some SNO modification cause loss of function in enzymes)?*

Follow-up comment: "Nitrosative stress" is different from peroxynitrite-mediated "nitro-oxidative" stress that leads to protein tyrosine nitration. The authors mixed-up these concepts. "Nitrosative stress" causes S-nitrosothiol formation, e.g. GSNO and S-nitrosylated proteins such as albumin, caspases, complex I, cyclophilin D and many more, that can be measured in plasma or tissues. So it would be appreciated if the authors could show markers of "nitrosative stress".

Regarding the figure on peroxynitrite formation that they show - I can neither find the figure nor the method description in the MS. How was it measured, please explain the method in very detail (just mentioning that it was a commercial kit will not suffice). The reason is that I am not aware of any specific method for detection of peroxynitrite and would be interested to know more about the presented assay.

Reply: We thank the reviewer for raising the important points mentioned above. Endogenous NO may cause S-nitrosylation (SNO), a type of posttranslational modification that can regulate protein function and cellular signalling. Although some

SNOs are involved in the pathogenesis of cardiovascular disease, there are still some SNOs showing positive effects. It has been accepted that the enhanced SNO of mitochondrial complex I inhibits its activity, thereby limiting ROS generation and providing cardioprotection³⁰⁻³². In this study, we also observed treatment with NO-patch increased the level of S-nitrosylated proteins in MI heart (Supplementary Fig. S15). Necessary discussion on this point has been included in the revised manuscript (Page 6, Lines 167-172).

In addition, the level of peroxynitrite (ONOO⁻) was detected according to the method we reported before [PMID: 26004323] as follows.

Briefly, one day after surgery, the rats were anesthetized by I.P. injection of 10% chloral hydrate (350 mg/kg), the hearts were collected and made into a homogenate. DAX-J2 PON Green at the concentration of 1 μ M was incubated with heart tissue homogenate (10 mg/mL) for 10 mins at 37 $^{\circ}$ C. Then, the fluorescent intensity of the reaction mixture was measured. Data were acquired through microplate reader with, λ_{ex} = 490 nm and λ_{em} = 525 nm.

8) *General comment: Whereas the presented data are impressive, the authors need to present a head-to-head comparison of protective effects of the nitrate-containing patch versus oral or intravenous organic nitrate administration versus inorganic nitrate treatment during ischemia/reperfusion. How is infarct size and cardiac function affected by these three treatment protocols in the same infarct model? There are also organic nitrates that are devoid of nitrate tolerance and therefore could be used at high dose (e.g. PETN, see PMID: 11527645, PMID: 24071762).*

Follow-up comment: Without the head-to-head comparison, the presented data on the protective effect of your NO-releasing patch against MI damage are almost useless because without the direct comparison to other treatments in the same MI damage model one cannot judge whether the patch method is superior or not. Please add the head-to-head comparison at least investigating some key parameters in the rat model.

Reply: We have added the head-to-head comparison between NO-patch, organic isosorbide mononitrate and inorganic sodium nitrate, the related results were shown in Supplementary Fig 18, and necessary discussion on this point has been included in the revised manuscript as follows (Page 13-14, Lines 370-388).

Furthermore, a head-to-head comparison study has been performed in order to demonstrate the advantages and disadvantages of nitrate-functionalized patches through systemic comparison with both organic nitrate, isosorbide mononitrate (iso-mono) and inorganic sodium nitrate (NaNO₃) that have been reported previously by utilising the rat cardiac ischemia/reperfusion(I/R) model. Ten days after treatment, cardiac histology and function were analyzed. Results show that after intake of sodium nitrate there was reduction in fibrotic area and infarct size (**Supplementary Fig 18a-c**). However, it didn't show a positive effect on the cardiac function (**Supplementary Fig 18d, e**). In contrast, treatment with isosorbide mononitrate and NO-patch significantly improved the outcomes of cardiac I/R injury, the fibrotic area as well as the infarct size were reduced (**Supplementary Fig 18a-c**), and the cardiac ejection function was improved (**Supplementary Fig 18d, e**). However, serious

cardiac arrest was observed in rats (3/7) after gavage of isosorbide mononitrate which could be caused by hypovolemic shock. As intake of isosorbide mononitrate resulted in whole-body vasodilation and hypotension, which turns to be the risk factor of hypovolemia. Besides, the amount of administrated iso-mono (0.15mmol) is higher than that of PCL-ONO₂ present in the NO-patch (0.25μmol). As a result, despite the comparable therapeutic efficacy, we believe that NO-patch implantation is a promising strategy for cardioprotection and cardiac repair in terms of biotransformation efficiency and safety due to the local NO generation that is responsive to ischaemia microenvironment after MI.

9) *General comment: Previously, it was shown that already improvement of organic nitrate bioactivation and prevention of nitrate tolerance may be sufficient to conserve the anti-ischemic effects and protection by organic nitrates such as nitroglycerin in a MI model (PMID: 18787169, PMID: 22049071). How would your nitrate-containing patch compete with this pharmacological activation of nitroglycerin bioactivating enzyme ALDH-2 by Alda-1?*

Follow-up comment: Answer almost fine. Please add a critical discussion on the advantage of your NO-releasing patch over the nitroglycerin/Alda-1 combination.

Reply: We thank the reviewer for this suggestion and have now included the relevant discussion in the revised manuscript as follows (Page12-13, Lines 344-352).

As a macromolecular nitrate, PCL-ONO₂ demonstrates a specific biotransformation pathway that is different from that of small molecule drugs. First, polymers undergo non-enzymatic hydrolysis to release nitrate anion locally in infarcted myocardium. Then, the generated nitrate anion is reduced to release NO via the nitrate-nitrite-NO sequential pathway. Since the phenomenon of tolerance is not exhibited with the consumption of inorganic nitrate/nitrite⁶¹, it is reasonable to speculate that PCL-ONO₂ developed in this study is free from tolerance because the process of biotransformation is different from that of classic organic nitrates. In addition, nitrate-functionalized PCL does not require additional activators (such as Alda-1), although relevant experiments are required to validate this hypothesis.

10) *General comment: I have seen much more impressive improvements by some treatment protocols in MI animal models in the past than those survival data presented in figure 6. Therefore, it is really important to compare head-to-head different treatment protocols (see my suggestions above, e.g. oral organic nitrate, inorganic nitrate, Alda-1) to demonstrate the advantage of the here presented method over those previously reported.*

Follow-up comment: see my new comment #8.

Reply: We have added the comparison study and the related discussion in the revised manuscript (Supplementary Fig 18 and Page 13-14, Lines 370-388).

11) *ok.*

12) ok.

13) ok.

14) ok.

15) *Other specific concerns: why was survival only tested in the pig model with very low amount of animals? I think with the rodents it would have been easier to have large animal numbers for the survival studies.*

Follow-up comment: Maybe in the future mice would be the better choice as they show appreciable mortality after LAD ligation induced MI. I still think that doing a survival study with less than 10 animals cannot yield very robust data.

Reply: I respectfully agree with you that mouse model is more appropriate for survival study after LAD ligation induced MI and it will be utilized in our further study.

16) ok.

17) ok.

18) ok.

Reviewer #4 (Remarks to the Author):

The revision addressed most of my concerns. However, the revision added a justification of the patch " Acellular cardiac patches are another therapy that treat MI by restricting ventricular dilatation, preventing adverse left ventricular remodelling and improving contractile function that is directly attributable to mechanical support by elastic polymers (Page 3, Lines 68-70) " This implies the author was claiming PCL was elastic. I don't think they claimed so, but this could lead to confusion. PCL is not an elastomer and I suggest the authors to add a line to clarify that.

Reply: We thank the reviewer for raising this important point, and the statement has been written in the revised manuscript as following (Page 3, Lines 68-70).

Acellular cardiac patches are another therapy that treat MI by restricting ventricular dilatation, preventing adverse left ventricular remodelling and improving contractile function that is directly attributable to the elastic (such as polyurethane) or viscoelastic properties^{22, 23}.

Reviewer #5 (Remarks to the Author):

The authors addressed most of the questions of the reviewers. The main concern of reviewer 2 is the novelty of the work and I strongly agree with him or her. The biodegradable polymeric NO donor used in this work was structurally similar to the reported ones and the mechanism of NO releasing was completely same. Although the authors emphasized the improved therapeutic efficacy, no theoretical reason was provided to explain why the polymer was better than the reported ones. On the basis of lack of theoretical support, it is difficult to judge the therapeutic efficacy with such a small sample amount.

Reply: We thank the reviewer for taking the time to carefully evaluate our manuscript. A range of additional experiments has been performed in order to illustrate the unique biotransformation mechanism that is different from organic small molecular nitrates as well as inorganic nitrate and provide theoretical support for the improved therapeutic efficacy. This resulted in addition of new figures (Figure 3, Supplementary Fig 9 and 11) as well as a new section in the text of the manuscript (Page 5, Lines 123-148). We also performed a head-to-head comparison study and have provided more detailed information on the advantages and disadvantages of nitrate-functionalized patches over both organic and inorganic nitrates that have been reported previously (Supplementary Fig 18 and Page 13-14, Lines 370-388), as requested by reviewer #3. Altogether, we believe that this has further strengthened the manuscript and hope that the reviewer will be now able to fully support the publication of our study.

REVIEWER COMMENTS

Reviewer #1 (Remarks to the Author):

Thank you for the revision, which answered my concerns and changed the message of the paper. It seems now that the NO-functionalized patch did not increase CM proliferation and confers benefit rather by a cardioprotective effect in the early stage of MI. This makes more sense. I have no further comments.

Reviewer #3 (Remarks to the Author):

Comments

1) ok.

2) ok.

3) Thank you for adding this detailed discussion on DES.

4) ok.

5) New response is fine. However, on page 10, do not use "Nitrates are found..." (and throughout the MS) as there is only one inorganic nitrate ion. So, please use "Nitrate is found..." and also correct elsewhere. In contrast, there are different organic nitrates. So you can use plural when talking about organic nitrates.

6) ok.

7) Additional experiments on S-nitrosylated proteins are appreciated. However, the authors write in the method on page 23 that the S-nitrosothiols were firstly reduced to the free thiols and then covalently bound to a fluorescein-coupled avidin reagent allowing fluorescent detection. How can the method then discriminate between SNO-groups and reduced thiols? Was in the description a step missing where all reduced thiols were alkylated, then remaining SNO-groups were reduced and then reacted with the fluorescein-coupled avidin reagent?

Regarding the peroxynitrite detection. The method and data are still not described in the MS. The mentioned paper PMID: 26004323 described NO detection but not peroxynitrite detection by DAX-J2 PON Green.

8) New data very much appreciated. So cardioprotective efficacy of the nitrate-patch is as good as by isosorbide mononitrate, however without the mentioned side effects such as hypovolemia.

9) ok.

10) ok.

11) ok.

12) ok.

13) ok.

14) ok.

15) ok.

16) ok.

17) ok.

18) ok.

Reviewer #5 (Remarks to the Author):

The manuscript has been well revised and I recommend the publication.

Reviewer Comments

Reviewer #1 (Remarks to the Author):

Thank you for the revision, which answered my concerns and changed the message of the paper. It seems now that the NO-functionalized patch did not increase CM proliferation and confers benefit rather by a cardioprotective effect in the early stage of MI. This makes more sense. I have no further comments.

Reply: We feel honored to receive such encouraging comments. Thanks very much for the support.

Reviewer #3 (Remarks to the Author):

1) *ok.*

2) *ok.*

3) *Thank you for adding this detailed discussion on DES.*

Reply: We thank the reviewer's valuable comment.

4) *ok.*

5) *New response is fine. However, on page 10, do not use "Nitrates are found..." (and throughout the MS) as there is only one inorganic nitrate ion. So, please use "Nitrate is found..." and also correct elsewhere. In contrast, there are different organic nitrates. So you can use plural when talking about organic nitrates.*

Reply: We have revised the manuscript thoroughly by adopting "Nitrate" and "Nitrates" to represent inorganic nitrate and organic nitrates, respectively.

6) *ok.*

7) *Additional experiments on S-nitrosylated proteins are appreciated. However, the authors write in the method on page 23 that the S-nitrosothiols were firstly reduced to the free thiols and then covalently bound to a fluorescein-coupled avidin reagent allowing fluorescent detection. How can the method then discriminate between SNO-groups and reduced thiols? Was in the description a step missing where all reduced thiols were alkylated, then remaining SNO-groups were reduced and then reacted with the fluorescein-coupled avidin reagent?*

Regarding the peroxynitrite detection. The method and data are still not described in the MS. The mentioned paper PMID: 26004323 described NO detection but not peroxynitrite detection by DAX-J2 PON Green.

Reply: We thank the reviewer for pointing this out. Actually, the starting step of this assay was incubating heart sections with blocking reagent, which blocked free thiols in the samples, and then followed by reduction and labeling with biotin as well as fluorescein-avidin conjugate, the intrinsic S-nitrosylated proteins can be detected. We

have attached the diagram of the assay as shown in user manual, and also revised the contents in the revised manuscript as follows (Page 23):

To proceed with this assay, heart sections were firstly incubated with blocking reagent to block free thiols in the samples, and then followed by reduction and labeling with maleimide-biotin as well as fluorescein-avidin, S-nitrosylated proteins can be localized.

Diagram of process for S-nitrosylated protein detection.

The method for the detection of peroxynitrite was described as follows. Since the related data was not present in the revised manuscript so it was removed accordingly. Briefly, one day after surgery, the rats were anesthetized by I.P. injection of 10% chloral hydrate (350 mg/kg), the hearts were collected and made into a homogenate. DAX-J2 PON Green (Amplite™ Fluorimetric Peroxynitrite Quantification Kit, AAT Bioquest, 16316) at the concentration of 1 μM was incubated with heart homogenate (10 mg/mL) for 10 min at 37 °C. Then, the fluorescent intensity of the reaction mixture was measured. Data were acquired through microplate reader with $\lambda_{ex} = 490$ nm and $\lambda_{em} = 525$ nm.

8) *New data very much appreciated. So cardioprotective efficacy of the nitrate-patch is as good as by isosorbide mononitrate, however without the mentioned side effects such as hypovolemia.*

Reply: The reviewer is correct. We thank the reviewer's suggestions in helping us to improve our manuscript.

9) *ok.*

10) *ok.*

11) *ok.*

12) *ok.*

13) *ok.*

14) *ok.*

15) *ok.*

16) *ok.*

17) *ok.*

18) *ok.*

Reviewer #5 (Remarks to the Author):

The manuscript has been well revised and I recommend the publication.

Reply: We thank this reviewer for the encouraging remarks and the positive evaluation.

REVIEWERS' COMMENTS

Reviewer #3 (Remarks to the Author):

All of my previous comments were addressed in full detail.